# HiPhO: How Far Are (M)LLMs from Humans in the Latest High School Physics Olympiad Benchmark?

## Abstract

Recently, the physical capabilities of (M)LLMs have garnered increasing attention. However, existing benchmarks for physics suffer from two major gaps: they neither provide systematic and up-to-date coverage of real-world physics competitions such as physics Olympiads, nor enable direct performance comparison with humans. To bridge these gaps, we present **HiPhO**, the first benchmark dedicated to high school physics Olympiads with human-aligned evaluation. Specifically, HiPhO highlights three key innovations. **(1) Comprehensive Data:** It compiles 13 latest Olympiad exams from 2024–2025, spanning both international and regional competitions, and covering mixed modalities that encompass problems spanning text-only to diagram-based. **(2) Professional Evaluation:** We adopt official marking schemes to perform fine-grained grading at both the answer and step level, fully aligned with human examiners to ensure high-quality and domain-specific evaluation. **(3) Comparison with Human Contestants:** We assign gold, silver, and bronze medals to models based on official medal thresholds, thereby enabling direct comparison between (M)LLMs and human contestants. Our large-scale evaluation of 30 state-of-the-art (M)LLMs shows that: across 13 exams, open-source MLLMs mostly remain at or below the bronze level; open-source LLMs show promising progress with multiple golds; closed-source reasoning MLLMs can achieve 6 to 12 gold medals; and most models still have a significant gap from full marks. These results highlight the performance gap between open-source models and top students, the strong reasoning abilities of closed-source models, and the remaining room for improvement. HiPhO, a human-aligned Olympiad benchmark for multimodal physical reasoning, is open-source at `https://anonymous.4open.science/r/HiPhO`.

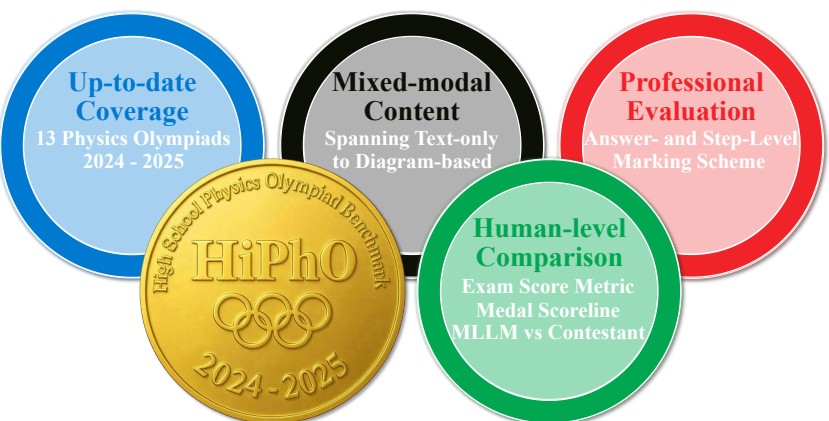

Figure 1: The overview of our HiPhO (High School Physics Olympiad) benchmark.

---

[2]In this work, medal thresholds are determined based on the theoretical exam scores of human medalists.

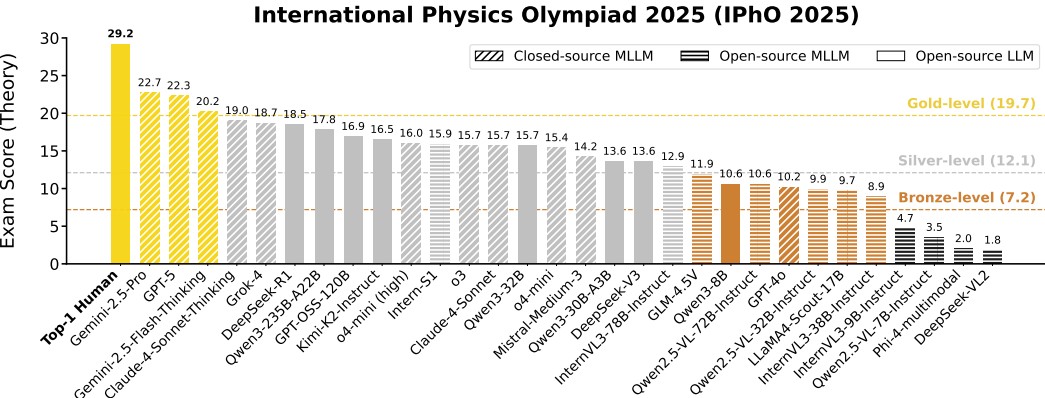

Figure 2: Model performance on the International Physics Olympiad 2025 (IPhO 2025)[2]. We show the performance gap between state-of-the-art (M)LLMs and the top-1 human contestant.

# 1 INTRODUCTION

> *"Physics is essentially an intuitive and concrete science."* — ALBERT EINSTEIN

Large language models (LLMs) and Multimodal LLMs (MLLMs) have recently attracted increasing attention for their physical reasoning capabilities. However, existing datasets for physics remain limited in scope, lacking both systematic coverage of physics Olympiads and direct comparison with human contestants. By contrast, mathematical Olympiads have already received extensive attention—models such as GPT-5 have achieved over 90% accuracy on challenging benchmarks like AIME-2025 (Ye et al., 2025), and DeepMind's Gemini ("Deep Think") even reached gold-medal level at the 2025 International Mathematical Olympiad (IMO). As the crown jewel of high school physics, Olympiad problems offer a uniquely rigorous testbed that remains underexplored, underscoring the need for a new benchmark with systematic evaluation and human-aligned comparison.

Physics Olympiads, such as the International Physics Olympiad (IPhO), represent the pinnacle of high school physics competitions. Unlike mathematical Olympiads, they require a deep understanding of real-world physical principles, the derivation of abstract physics formulas, and the ability to reason with complex multimodal diagrams. These characteristics make physics Olympiads an ideal testbed for evaluating whether (M)LLMs can perform authentic visual and physical reasoning. Moreover, they offer the unique advantage of enabling direct comparison with human contestants.

However, current Olympiad-related physics datasets exhibit several critical limitations: **(1) Outdated coverage:** Datasets like OlympiadBench (He et al., 2024) includes IPhO problems only up to 2021 and Asian Physics Olympiad (APhO) problems up to 2015, omitting the most recent two years of exams. **(2) Lack of multimodal content:** Physics Olympiad problems often involve complex diagrams. However, datasets such as PHYBench (Qiu et al., 2025) and PHYSICS (Zheng et al., 2025) are entirely text-based. **(3) Limited evaluation quality:** Most datasets stop at coarse answer-level evaluation, without step-level grading aligned with official marking schemes. **(4) No human-level comparison:** Existing datasets typically report accuracy, without exam-score-based comparisons against human contestants in real-world physics competitions.

To address these limitations, we introduce **HIPHO** (High School Physics Olympiad), the first benchmark dedicated to recent physics Olympiads with human-aligned evaluation. As illustrated in Fig. 1, HIPHO incorporates four key improvements: **(1) Up-to-date coverage:** It compiles the latest 13 Olympiad exams from 2024–2025, including both international and regional contests. **(2) Mixed-modal content:** Problems are collected as complete exams, spanning text-only to diagram-based, with fine-grained categorization of diagram types (see Fig. 4). **(3) Professional evaluation:** We adopt official marking schemes to perform fine-grained grading at both the answer and step level, fully aligned with human examiners to ensure rigorous and domain-specific assessment. **(4) Human-level comparison:** We compute full exam scores for models and map them to gold, silver, and bronze medal thresholds, thereby enabling direct comparison with human contestants.

In our large-scale evaluation of 30 state-of-the-art (M)LLMs, we observe a clear performance hierarchy. **Closed-source reasoning MLLMs** dominate the medal table, winning 6–12 gold medals in

Table 1: Comparison of physics benchmarks across key dimensions, including Olympiad coverage, modalities, evaluation, metrics, and use of medal scores. HIPHO introduces three innovations: (1) fine-grained modality categorization: 1 = text-only, 2 = text + illustration figure, 3 = text + variable figure, 4 = text + data figure (see Fig. 4); (2) step-level grading using official marking schemes; and (3) the first benchmark to compare model performance with medal cutoffs and contestant scores.

| Benchmark | Multimodal | Modality Type | Evaluation Method | Performance Metric | Medal Score |
|---|---|---|---|---|---|
| *Non-Olympiad Physics Benchmarks or Subsets* | | | | | |
| GPQA (Rein et al., 2024) | ✗ | 1 | Answer | Acc. | ✗ |
| HLE (Phan et al., 2025) | ✗ | 1 | Answer | Acc. | ✗ |
| UGPhysics (Xu et al., 2025) | ✗ | 1 | Answer | Acc. | ✗ |
| CMPhysBench (Wang et al., 2025b) | ✗ | 1 | Answer | Acc., SEED Score | ✗ |
| PhyX (Shen et al., 2025) | ✓ | 2 | Answer | Acc. | ✗ |
| PhysUniBench (Wang et al., 2025a) | ✓ | 2 | Answer | Acc. | ✗ |
| *Olympiad-related Physics Benchmarks or Subsets* | | | | | |
| PHYBench (Qiu et al., 2025) | ✗ | 1 | Answer | Acc., EED Score | ✗ |
| PHYSICS (Zheng et al., 2025) | ✗ | 1 | Answer | Acc. | ✗ |
| OlympiadBench (He et al., 2024) | ✓ | 1,2 | Answer | Acc. | ✗ |
| OlympicArena (Huang et al., 2024) | ✓ | 1,2 | Answer+Model Step[1] | Acc. | ✗ |
| PhysReason (Zhang et al., 2025) | ✓ | 1,2 | Answer+Error Step[2] | Acc. | ✗ |
| SeePhys (Xiang et al., 2025) | ✓ | 1,2,4 | Answer | Acc. | ✗ |
| *Olympiad-focused Physics Benchmark* | | | | | |
| **HIPHO (Ours)** | ✓ | 1,2,3,4 | Answer+Marking Step[3] | Exam Score | ✓ |

[1]GPT4-generated steps; [2]First-error-step detection in model output; [3]Official marking scheme with scoring per predefined point.

13 Olympiads. However, even the strongest models, such as Gemini-2.5-Pro and GPT-5, still fall short of the very best human contestants, especially in challenging exams like IPhO and EuPhO. In contrast, **open-source chat MLLMs** failed to secure any gold medals, with most scoring only at or below the bronze threshold. More encouragingly, several **open-source reasoning (M)LLMs**, including Intern-S1 and DeepSeek-R1, each achieved 4–8 gold medals, particularly in relatively easier exams such as F=MA. Taken together, these results reveal the strong but still non-parity performance of closed-source models, the evident limitations of open-source chat MLLMs, and the promising trajectory of open-source reasoning (M)LLMs in advancing physics problem solving.

In summary, our main contributions are as follows:

• **Olympiad-focused Benchmark.** We present HIPHO, the first benchmark dedicated to high school physics Olympiads, comprising 13 Olympiad exam papers from 2024–2025. All problems are expert-curated, structurally extracted, and manually verified to ensure high quality and consistency.

• **Human-aligned Scoring.** We evaluate performance using exam scores rather than the commonly used accuracy, applying both answer- and step-level grading based on official marking schemes. This enables direct, medal-standard comparisons with actual human contestants.

• **Human-level Comparison.** Compared to human contestants, closed-source reasoning MLLMs reach gold in 6–12 exams, while open-source MLLMs stay mostly at bronze; some open-source LLMs show stronger reasoning with multiple golds, yet all remain far from the very top students.

## 2 RELATED WORK

Among Olympiad-related physics datasets (Table 1), *PHYBench* (Qiu et al., 2025) and *PHYSICS* (Zheng et al., 2025) are text-only, while *OlympiadBench* (He et al., 2024) and *OlympicArena* (Huang et al., 2024) cover outdated exams. Multimodal datasets such as *SeePhys* (Xiang et al., 2025) and *PhysReason* (Zhang et al., 2025) remain restricted in scope (e.g., IPhO or CPhO). By contrast, HIPHO compiles 13 recent Olympiads (2024–2025), spanning problems with mixed modalities. More importantly, whereas existing datasets generally stop at coarse answer-level accuracy, HIPHO introduces fine-grained evaluation based on official marking schemes and adopts exam scores as the metric, enabling direct comparison between (M)LLMs and human contestants.

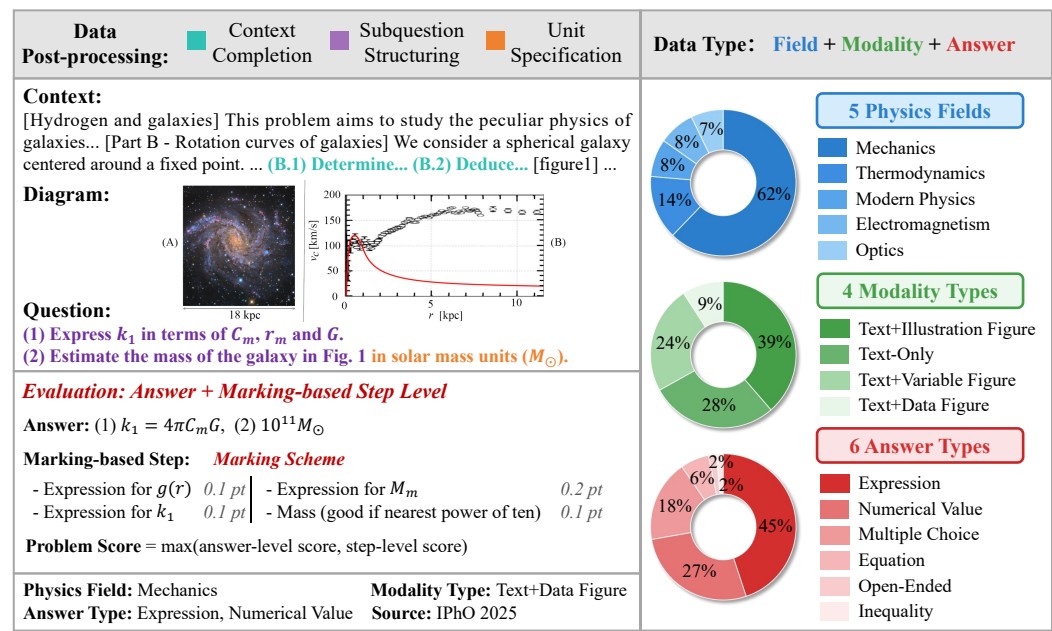

Figure 3: Framework and statistics of the HIPHO benchmark. **Top left:** An example problem with three post-processing steps—context completion, subquestion structuring, and unit specification. **Bottom left:** Answer + step-level evaluation based on official marking schemes. **Right:** Dataset composition across five physics fields, four modality types, and six answer types.

## 3 THE HIPHO BENCHMARK FOR HIGH SCHOOL PHYSICS OLYMPIAD

### 3.1 OVERVIEW

We introduce HIPHO, the first high school physics Olympiad benchmark designed to compare the reasoning capabilities of (M)LLMs against human contestants. It contributes along three key dimensions: (1) a comprehensive and up-to-date dataset, (2) professional evaluation aligned with official marking schemes, and (3) human-level comparison.

**Dataset Perspective.** As shown in Table 2, HIPHO covers 360 problems and 519 subquestions from 13 Olympiad exams in 2024–2025, making it the most up-to-date benchmark. Each problem is categorized along two axes: **(1) physics taxonomy**—five fields (Mechanics, Electromagnetism, Thermodynamics, Optics, Modern Physics); and **(2) modality type**—four formats (Text-Only, Text+Illustration Figure, Text+Variable Figure, Text+Data Figure; see Fig. 4), enabling detailed analysis of physical and visual reasoning.

Table 2: Benchmark Statistics.

| Statistic | Count |
|---|---|
| *Data Level* | |
| Total Problems | 360 |
| - Total Subquestions | 519 |
| Total Physics Fields | 5 |
| Total Modality Types | 4 |
| Total Answer Types | 6 |
| Total Difficulty Levels | 3 |
| Language (EN : ZH) | 308:52 |
| *Evaluation Level* | |
| Total Exam Papers | 13 |
| - With Medal Scorelines | 13 |
| - With Marking Schemes | 7 |

**Evaluation Perspective.** HIPHO introduces the first evaluation method that integrates answer-level correctness with step-level assessment using *official marking schemes*. In addition, we report *exam scores* rather than accuracy to more faithfully reflect overall performance. This allows direct comparison with human scores and medal thresholds, providing the first quantitative analysis of the gap between state-of-the-art (M)LLMs and top human contestants in real-world competitions.

### 3.2 DATASET CONSTRUCTION

**Up-to-date Coverage.** HIPHO includes 13 exams from seven major Physics Olympiads (PhOs) and competitions in 2024–2025 (Appendix B.1). Selection was based on influence and the availability

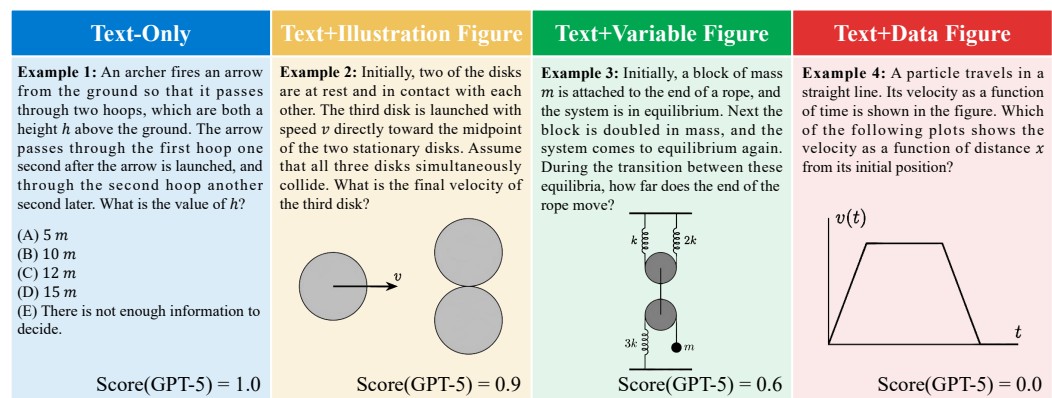

Figure 4: Examples of the four modality types in the HɪPʜO benchmark: Text-Only (TO), Text+Illustration Figure (TI), Text+Variable Figure (TV), and Text+Data Figure (TD). TO and TI rely mainly on text, whereas TV and TD require understanding variables or data, making them more challenging for visual reasoning, as reflected in GPT-5's scores (each example worth 1.0 point).

of human scores[3]. With data processing tools, the benchmark can be efficiently updated each year, ensuring continuously refreshed coverage. We further group them into three difficulty levels:

- **Hard:** *IPhO* (International PhO), *APhO* (Asian PhO), *EuPhO* (European PhO)

- **Medium:** *NBPhO* (Nordic-Baltic PhO), *PanPhO* (Pan Pearl River Delta PhO)

- **Easy:** *PanMechanics* (Pan Pearl River Delta Mechanics Test), *F=MA*

**High-Quality Extraction Pipeline.** We collect PDFs from official sources[4] and process them through a structured pipeline: **(1) Data Extraction:** PDFs are converted to markdown via OCR, preserving LaTeX expressions. **(2) QA Matching:** Question indices align each problem with its answer. **(3) Human Verification:** Every QA pair is manually checked—experts fix OCR errors, resolve mismatches, and validate answers, ensuring full fidelity to the original exams. **(4) Marking Scheme Structuring:** Where available, step-level criteria are extracted and reformatted into model-readable rubrics for fine-grained evaluation. **(5) Post-processing:** Verified QAs are refined for context, question, and unit clarity, as detailed below.

**Post-Processing and QA Refinement.** To better simulate exam conditions and improve evaluation accuracy, we apply three post-processing steps (see Fig. 3). **(1) Context Completion:** Long stems with interdependent subquestions are merged to ensure coherence. **(2) Subquestion Structuring:** Semantically linked subquestions are preserved and explicitly labeled (e.g., "Find the speed and acceleration" → "(1) Find the speed. (2) Find the acceleration") to guide ordered responses and reduce omissions. **(3) Unit Specification:** Required units are made explicit (e.g., "Find the speed in $m/s$") to prevent trivial mismatches, ensuring fairness without changing problem difficulty.

**Comprehensive Data Annotation.** To enable fine-grained analysis, we annotate the dataset along three dimensions. **(1) Physics Field:** Problems are labeled by five fields—Mechanics, Electromagnetism, Thermodynamics, Optics, and Modern Physics. **(2) Visual Modality:** As shown in Fig. 4, problems are categorized into four types, reflecting varying levels of visual reasoning:

- *Text-Only (TO)*: Problems are stated entirely in text with no figures.

- *Text+Illustration Figure (TI)*: Figures depict scenarios, while the text provides the description.

- *Text+Variable Figure (TV)*: Figures specify key variables or spatial configurations.

- *Text+Data Figure (TD)*: Figures present data, plots, or functions that are not given in the text.

---

[3]Notably, CPhO, USAPhO, and APhO-2024 were excluded due to the absence of human scores.

[4]E.g., IPhO 2024–2025: `https://ipho.olimpicos.net/`; other sources in Appendix B.1.

**(3) Answer Type:** Subquestions are classified into six types: Expression, Numerical Value, Multiple Choice, Equation, Inequality, and Open-Ended. As shown in Fig. 3, *Expression* (45%) and *Numerical Value* (27%) dominate, assessing models' ability in symbolic reasoning and precise calculation.

### 3.3 EVALUATION METHOD

**Comprehensive Evaluation Framework.** Our framework combines *answer-level* and *step-level* scoring, closely mirroring human grading. **(1) Answer-level (coarse):** A rule-based math-verifier (Kydlíček) checks whether the answer matches the ground truth; if the check fails, possibly due to equivalent expressions, we invoke a strong model to assess correctness. **(2) Step-level (fine):** Model solutions are compared with official marking schemes, with partial credit awarded for correct steps. The score of each problem is taken as the maximum of its answer-level and step-level scores, ensuring full credit for correct answers and fair grading for intermediate steps. For problems with multiple official solutions (e.g., NBPhO), all schemes are scored and the best result taken. Exam scores are then obtained by summing over problems.

**Key Improvements in Evaluation.** Our evaluation introduces three major improvements. **(1) Step-level scoring:** Unlike most datasets that grade only final answers (Table 1), we recognize partial correctness through step-level scoring. **(2) Marking-scheme alignment:** We are the **first** to extract steps directly from official marking schemes, awarding points by explicit criteria for rigor and consistency, while OlympicArena generates steps with GPT-4. **(3) Stronger grader:** We adopt Gemini-2.5-Flash, while prior works (Xu et al., 2025; Feng et al., 2025) rely on GPT-4o, whose lower competition scores make it less suitable for Olympiad-level evaluation. Gemini-2.5-Flash yields exam scores closer to human experts (within 1 point in IPhO 2025 and 2024; Appendix E.3).

### 3.4 DATA LEAKAGE

To access the model's memorization on benchmark data, we conducted explicit data leakage checks using the **partial-prompt completion** strategy (Wu et al., 2025). We randomly sampled 90 problems (25% of the benchmark), ensuring balanced coverage across all Olympiads. For each sampled problem, we provided the first 60% of the content and asked the model to greedily complete the remainder. We then evaluated the outputs using the same metrics as in Wu et al. (2025): **(1) ROUGE-L (Partial-Prompt Completion Rate)** (Lin, 2004), which measures the longest common subsequence overlap between generated and reference text, and **(2) ACC (Partial-Prompt Answer Accuracy)**, which measures whether the generated continuation contains the correct answer.

As reported in Appendix B.3, Table 7 show that partial-prompt completion rates across 10 top-performing models are consistently below 15%, with answer accuracy below 4% on average. The per-Olympiad breakdowns in Tables 8 and 9 further show that across all 13 Olympiads, most completion rates remain under 20%, and most answer accuracies remain at 0%. These results confirm that the HiPhO benchmark exhibits minimal data leakage and remains highly challenging, as models fail to reproduce or solve most potentially leaked instances.

## 4 EXPERIMENTS

To evaluate state-of-the-art (M)LLMs against top Olympiad contestants, we tested 30 representative models: 11 closed-source MLLMs, 11 open-source MLLMs, and 8 open-source LLMs. Details of the setup and model categorization are in Appendix D.1, and medal scorelines are in Appendix D.2.

### 4.1 MAIN RESULTS

**What are the most powerful (M)LLMs in physics Olympiads?** The medal table in Table 3 ranks models just like an Olympiad. The top five positions are occupied by **Gemini-2.5-Pro**, **Gemini-2.5-Flash-Thinking**, **GPT-5**, **o3**, and **Grok-4**. However, in the challenging IPhO-2025, only three of them reached the gold threshold, with the top two separated by merely 0.4 points. This narrow margin highlights the intense competition at the frontier of closed-source model performance.

**How far are (M)LLMs from top-performing students?** HIPHO enables human-level comparisons. **(1) Closed-source reasoning MLLMs** collected 6–12 gold medals across the 13 exams, yet still fell short of the very best students—for instance, in IPhO-2025 the top human scored **29.2/30**, while the best model achieved only **22.7/29.4**. **(2) Open-source MLLMs** mostly remained at or below the bronze threshold, with Intern-S1 the only exception to reach gold. **(3) Open-source LLMs** generally outperformed open-source MLLMs, earning gold in easier contests such as F=MA and even reaching the gold threshold in IPhO-2024, but still lagged well behind the very top students.

Table 3: Evaluation results on the HiPhO benchmark (13 physics Olympiads from 2024–2025) using the *exam score* metric. Gold, Silver and Bronze indicate scores above the respective thresholds. Models are ranked by medal counts; **bold** is the highest score, and underline is the second highest. Here, only the theoretical parts of exams are used, hence Full Mark (Model) ≤ Full Mark (Human).

| Physics Olympiad | IPhO | | APhO | EuPhO | | NBPhO | | PanPhO | | PanMechanics | | F=MA | | Medal Table | | |
|---|---|---|---|---|---|---|---|---|---|---|---|---|---|---|---|---|
| Year | 2025 | 2024 | 2025 | 2025 | 2024 | 2025 | 2024 | 2025 | 2024 | 2025 | 2024 | 2025 | 2024 | | | |
| Full Mark (Human) | 30.0 | 30.0 | 30.0 | 30.0 | 30.0 | 72.0 | 72.0 | 100.0 | 100.0 | 100.0 | 100.0 | 25.0 | 25.0 | | | |
| Full Mark (Model) | 29.4 | 29.3 | 30.0 | 29.0 | 28.0 | 43.5 | 50.0 | 100.0 | 98.0 | 100.0 | 100.0 | 25.0 | 25.0 | | | |
| Top-1 Score (Human) | 29.2 | 29.4 | 30.0 | 27.0 | 30.0 | 53.2 | 40.8 | 81.0 | 66.5 | 62.0 | 51.0 | 25.0 | 24.0 | | | |
| Top-1 Score (Model) | 22.7 | 25.9 | 27.9 | 14.9 | 21.9 | 34.1 | 35.9 | 60.3 | 75.4 | 72.1 | 79.0 | 22.8 | 22.4 | | | |
| Gold Medal | 19.7 | 20.8 | 23.3 | 16.5 | 20.4 | 28.6 | 26.5 | 41.5 | 52.0 | 52.0 | 51.0 | 15.0 | 14.0 | ♂ | | |
| Silver Medal | 12.1 | 11.1 | 18.7 | 9.8 | 14.2 | 20.1 | 19.4 | 28.5 | 37.5 | 36.0 | 26.0 | 11.0 | 12.0 | | ♂ | |
| Bronze Medal | 7.2 | 3.6 | 13.1 | 5.8 | 8.9 | 15.2 | 13.5 | 14.5 | 16.0 | 20.0 | 12.0 | 9.0 | 10.0 | | | ♂ |
| *Closed-Source Reasoning MLLMs* | | | | | | | | | | | | | | | | |
| Gemini-2.5-Pro | **22.7** | **25.9** | **27.9** | **14.9** | 21.8 | 32.3 | **35.9** | **60.3** | 64.1 | 69.5 | 70.2 | **22.8** | 22.0 | 12 | 1 | 0 |
| Gemini-2.5-Flash-Thinking | 20.2 | 23.9 | 27.4 | 13.2 | 21.9 | 29.0 | 29.3 | 44.6 | 54.9 | 60.5 | 55.9 | 17.8 | 19.1 | 12 | 1 | 0 |
| GPT-5 | 22.3 | 20.2 | 27.0 | 10.3 | 21.7 | 32.9 | 32.8 | 55.9 | 69.8 | 69.4 | 79.0 | 22.4 | 22.4 | 11 | 2 | 0 |
| o3 | 15.7 | 23.7 | 25.9 | 11.4 | 21.6 | 34.1 | 33.5 | 47.3 | 55.9 | 71.4 | 75.6 | 22.0 | 20.6 | 11 | 2 | 0 |
| Grok-4 | 18.7 | 23.5 | 25.0 | 11.5 | 20.5 | 25.8 | 29.3 | 45.0 | 75.4 | 72.1 | 78.6 | 19.8 | 19.8 | 10 | 3 | 0 |
| Claude-4-Sonnet-Thinking | 19.0 | 22.0 | 24.8 | 9.7 | 20.5 | 28.1 | 25.6 | 43.1 | 39.3 | 57.4 | 61.8 | 19.2 | 20.1 | 8 | 4 | 1 |
| o4-mini | 15.4 | 22.9 | 22.8 | 10.1 | 20.9 | 26.9 | 27.3 | 39.4 | 47.1 | 64.2 | 62.5 | 18.6 | 18.5 | 7 | 6 | 0 |
| o4-mini (high) | 16.0 | 23.7 | 22.9 | 12.0 | 20.1 | 27.4 | 29.8 | 41.4 | 50.9 | 69.1 | 67.3 | 18.6 | 18.8 | 6 | 7 | 0 |
| *Open-Source Reasoning MLLMs* | | | | | | | | | | | | | | | | |
| Intern-S1 | 15.9 | 14.2 | 21.7 | 9.0 | 16.6 | 23.0 | 20.5 | 41.1 | 50.3 | 60.4 | 57.4 | 18.4 | 19.5 | 4 | 8 | 1 |
| GLM-4.5V | 11.9 | 4.4 | 16.2 | 8.7 | 14.1 | 19.5 | 14.0 | 18.5 | 16.0 | 47.8 | 39.0 | 13.0 | 13.8 | 0 | 4 | 9 |
| *Closed-Source Chat MLLMs* | | | | | | | | | | | | | | | | |
| Claude-4-Sonnet | 15.7 | 19.2 | 22.8 | 9.5 | 16.5 | 27.5 | 21.3 | 40.4 | 43.3 | 46.5 | 48.5 | 16.8 | 16.5 | 2 | 10 | 1 |
| Mistral-Medium-3 | 14.2 | 14.1 | 19.9 | 8.5 | 12.2 | 20.4 | 19.6 | 30.8 | 28.6 | 32.9 | 36.1 | 13.9 | 14.1 | 1 | 8 | 4 |
| GPT-4o | 10.2 | 9.4 | 15.1 | 6.8 | 9.2 | 16.4 | 11.7 | 27.8 | 22.8 | 28.2 | 26.5 | 15.0 | 10.9 | 1 | 1 | 10 |
| *Open-Source Chat MLLMs* | | | | | | | | | | | | | | | | |
| InternVL3-78B-Instruct | 12.9 | 12.5 | 17.7 | 7.5 | 15.2 | 22.3 | 22.5 | 26.2 | 27.4 | 21.1 | 27.1 | 12.0 | 13.0 | 0 | 8 | 5 |
| Qwen2.5-VL-72B-Instruct | 10.6 | 7.2 | 13.6 | 6.1 | 8.1 | 13.3 | 11.4 | 26.8 | 18.2 | 24.0 | 28.5 | 13.5 | 9.8 | 0 | 2 | 7 |
| LLaMA4-Scout-17B | 9.7 | 9.5 | 13.1 | 5.4 | 10.4 | 22.8 | 12.8 | 26.6 | 24.1 | 35.4 | 34.5 | 8.1 | 6.4 | 0 | 2 | 7 |
| Qwen2.5-VL-32B-Instruct | 9.9 | 8.2 | 16.5 | 6.9 | 10.0 | 15.3 | 14.4 | 22.5 | 22.4 | 28.1 | 29.9 | 7.6 | 4.6 | 0 | 1 | 10 |
| InternVL3-38B-Instruct | 8.9 | 7.8 | 12.3 | 6.1 | 8.3 | 14.0 | 10.6 | 24.1 | 20.4 | 27.5 | 24.8 | 8.2 | 6.8 | 0 | 0 | 7 |
| InternVL3-9B-Instruct | 4.7 | 3.7 | 7.2 | 4.2 | 4.1 | 9.4 | 6.0 | 12.4 | 11.0 | 11.6 | 16.3 | 6.4 | 6.2 | 0 | 0 | 2 |
| Qwen2.5-VL-7B-Instruct | 3.5 | 2.5 | 5.7 | 4.4 | 3.6 | 7.3 | 5.5 | 14.7 | 7.6 | 9.8 | 11.5 | 4.4 | 3.5 | 0 | 0 | 1 |
| Phi-4-multimodal | 2.0 | 1.6 | 4.2 | 3.6 | 3.6 | 5.0 | 4.5 | 8.3 | 9.0 | 10.0 | 10.1 | 4.4 | 5.0 | 0 | 0 | 0 |
| DeepSeek-VL2 | 1.8 | 0.5 | 2.5 | 3.4 | 3.4 | 5.0 | 3.2 | 5.6 | 4.8 | 7.3 | 6.4 | 5.0 | 3.9 | 0 | 0 | 0 |
| *Open-Source LLMs* | | | | | | | | | | | | | | | | |
| DeepSeek-R1 | 18.5 | 24.6 | 25.4 | 10.8 | 21.4 | 26.3 | 20.5 | 42.2 | 47.4 | 65.4 | 72.5 | 18.3 | 18.5 | 8 | 5 | 0 |
| Qwen3-235B-A22B | 17.8 | 23.8 | 26.0 | 9.2 | 21.5 | 28.4 | 31.1 | 42.3 | 49.1 | 63.1 | 44.6 | 18.4 | 17.4 | 8 | 4 | 1 |
| Qwen3-32B | 15.7 | 19.3 | 23.9 | 9.8 | 21.2 | 28.9 | 24.1 | 36.6 | 41.8 | 67.0 | 59.2 | 18.9 | 16.6 | 7 | 6 | 0 |
| Kimi-K2-Instruct | 16.5 | 19.8 | 24.2 | 11.0 | 16.9 | 26.5 | 26.2 | 35.9 | 41.8 | 65.9 | 58.9 | 16.0 | 18.2 | 5 | 8 | 0 |
| GPT-OSS-120B | 16.9 | 21.4 | 22.8 | 9.1 | 19.9 | 26.0 | 25.8 | 37.4 | 41.8 | 57.1 | 59.7 | 17.8 | 17.6 | 5 | 7 | 1 |
| Qwen3-30B-A3B | 13.6 | 15.4 | 22.7 | 9.8 | 16.5 | 24.7 | 21.7 | 31.9 | 39.5 | 49.9 | 45.0 | 15.5 | 15.0 | 2 | 11 | 0 |
| DeepSeek-V3 | 13.6 | 16.4 | 22.1 | 7.1 | 17.2 | 21.1 | 17.3 | 37.2 | 35.0 | 48.4 | 46.5 | 14.1 | 15.6 | 1 | 9 | 3 |
| Qwen3-8B | 10.6 | 12.7 | 11.5 | 7.1 | 11.9 | 20.1 | 17.3 | 26.3 | 22.3 | 21.8 | 22.8 | 10.8 | 10.0 | 0 | 2 | 10 |

**Can open-source models catch up with closed-source champions?** While closed-source reasoning models still dominate, recent progress in the open-source community is noteworthy. Intern-S1 distinguished itself as the only open-source MLLM to win gold, obtaining four medals. Even more impressively, open-source LLMs including DeepSeek-R1 and Qwen3-235B-A22B both achieved eight gold medals. These advances reveal both the enduring gap with closed-source leaders and the emerging potential of open-source research to narrow it.

## 4.2    CAN SOTA (M)LLMs COMPETE WITH HUMAN MEDALISTS IN PHYSICS OLYMPIADS?

**SOTA closed-source MLLMs remain unable to rival top-1 human contestants in most exams.** As shown in Fig. 5, closed-source MLLMs frequently reached the gold threshold across Olympiads, yet consistently fell short of the very best human medalists in most exams. For example, Gemini-2.5-Pro achieved 22.7 points in IPhO-2025, the highest among all models, but 21 of the 37 human gold medalists still scored higher. In PanPhO-2025, its 60.3 points were far below the top human score of 81. These results show that while closed-source models can reliably "reach the podium," they remain unable to rival the very top-performing students.

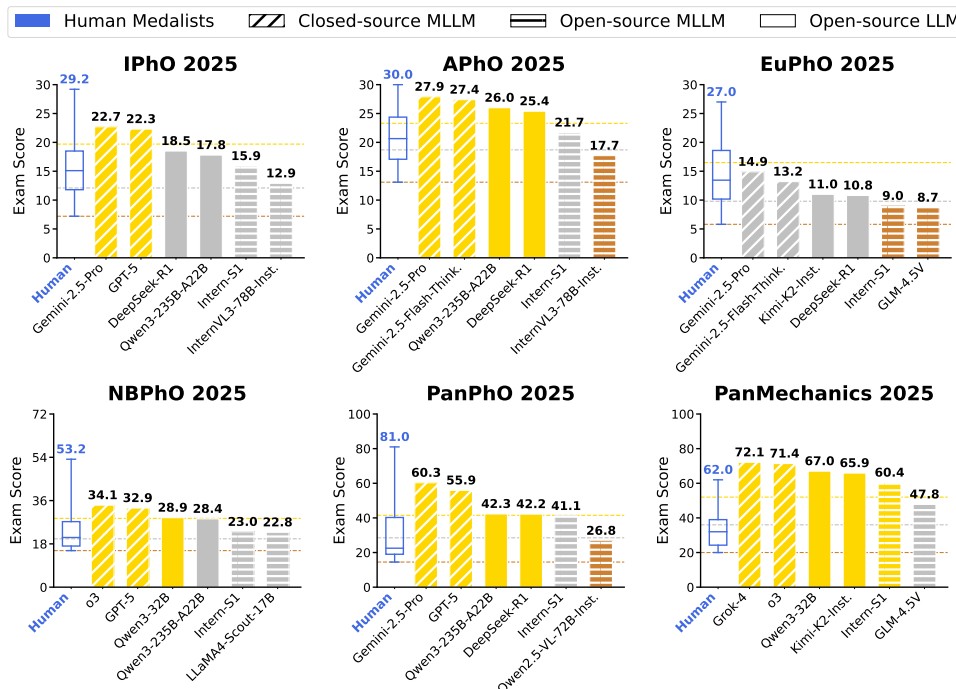

Figure 5: Exam scores of SOTA (M)LLMs and top-performing human medalists on the 2025 physics Olympiads. Each panel shows the score distribution of **human medalists** (boxplot), alongside bars for the top-2 models in closed-source MLLMs, open-source MLLMs, and open-source LLMs. Bars shaded Gold, Silver and Bronze indicate scores above their respective medal thresholds. As shown, SOTA (M)LLMs often reach gold level but still fall short of the highest human scores in most exams.

**SOTA open-source MLLMs usually reach bronze to silver levels, but still fall short of human gold medalists in most Olympiads.** As shown in Fig. 5, Intern-S1 is the strongest open-source MLLM but only secures 1 gold, 4 silver, and 1 bronze across six Olympiads, with the single gold from the relatively easy PanMechanics. On harder exams such as EuPhO, Intern-S1 lags far behind human medalists, achieving only a bronze-level score. Other open-source MLLMs, including GLM-4.5V and Qwen2.5-VL-72B-Instruct, perform even lower, rarely surpassing the silver threshold. These results highlight the persistent gap between open-source MLLMs and human gold medalists.

**SOTA open-source LLMs achieve multiple gold medals and generally surpass the median of human medalists.** As shown in Table 3, open-source LLMs such as DeepSeek-R1 and Qwen3-235B-A22B each achieve 8 golds across 13 Olympiads, far exceeding the performance of open-source MLLMs and even surpassing closed-source models like o4-mini. While these LLMs lack multimodal perception and thus struggle on vision-dependent problems, they nonetheless demonstrate strong reasoning ability in text-dominant tasks. As shown in Fig. 5, SOTA LLMs' scores exceed the median of human medalists in five of six Olympiads, with EuPhO-2025 as the only exception, highlighting their competitiveness in Olympiad-level physics reasoning.

## 5 DISCUSSION

### 5.1 ANALYSIS OF VISUAL MODALITY CHALLENGES IN MLLMS' PHYSICS REASONING

To investigate the impact of modality on performance, we categorize all problems into four types (Fig. 4): Text-Only (TO), Text+Illustration Figure (TI), Text+Variable Figure (TV), and Text+Data Figure (TD). Because total points per problem vary widely across Olympiads, rescaling scores introduces bias by overweighting high-value questions. To ensure fair comparison, we report the Mean Normalized Score (MNS) to avoid this bias by treating all problems equally:

$$\text{MNS}(M) = \frac{1}{N_M} \sum_{Q \in M} \frac{\text{Exam Score}(Q)}{\text{Full Mark}(Q)} \times 100\%, \quad (1)$$

where $M \in \{\text{TO}, \text{TI}, \text{TV}, \text{TD}\}$, $N_M$ is the number of questions in $M$, and $Q$ is a single question.

As shown in Fig. 6, diagram-based problems are consistently more difficult than text-only ones, leading to a sharp decline in mean normalized scores. For instance, the leading closed-source model, Gemini-2.5-Pro, the score reaches 86% on TO but decreases progressively as visual complexity increases: 81% on TI, 75% on TV, and 67% on TD. Grok-4 drops to 51% on TD, highlighting persistent challenges in extracting numerical values from data figures. Comparatively, open-source models fall further behind: the gap is visible on TO questions and widens significantly with visual input. On TV problems, GPT-5 achieves 75%, while Qwen2.5-VL-72B-Instruct reaches only 29%, showing its limitation in interpreting figures with complex variables.

Overall, these results highlight three major areas where the visual reasoning of MLLMs in physics problems can be further advanced: (**1**) interpreting illustration diagrams, (**2**) reasoning over variable-based graphs, and (**3**) accurately extracting quantitative information from data figures.

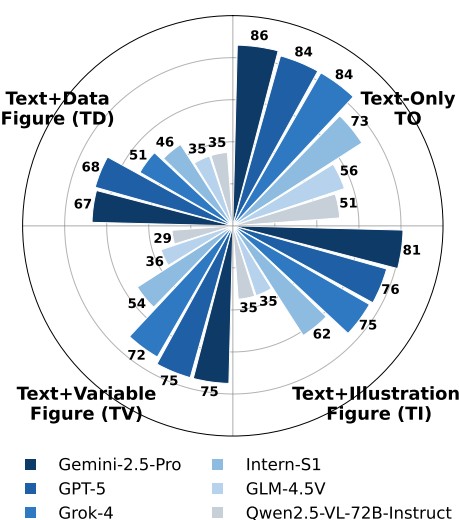

Figure 6: Comparison of mean normalized scores across four modality types.

## 5.2 ANALYSIS OF PHYSICS FIELD CHALLENGES FOR MLLMs

Physics Olympiad problems span five fields: **Mechanics** (Mech.), **Electromagnetism** (Elec.), **Thermodynamics** (Ther.), **Optics** (Opt.), and **Modern Physics** (Mode.). We report mean normalized scores (Eq. 1) for each field in Table 4, leading to three key observations. (**1**) **Optics is the most challenging field**, with all models scoring below 55%. Its difficulty arises from two aspects: geometrical optics requires diagram interpretation, while wave optics demands symbolic derivations—both remain weak points for current MLLMs. (**2**) **Modern Physics tends to yield higher scores than Mechanics**, likely because the problems are less visually intensive and the concepts tested do not extend to university-level depth. (**3**) **Mechanics and Electromagnetism show a clear source gap.** Closed-source models average above 70%. Most open-source MLLMs remain below 45%, with one exception: Intern-S1 narrows the gap, reaching 63–67% in Mech./Elec. Overall, optics is the most challenging, while closed-source models show steady gains.

Table 4: Comparison of mean normalized scores (%) across five major physics fields.

| Physics Field | Mech. | Elec. | Ther. | Opt. | Mode. |
|---|---|---|---|---|---|
| Gemini-2.5-Pro | 80 | 79 | 89 | 55 | 84 |
| GPT-5 | 80 | 72 | 82 | 52 | 79 |
| Grok-4 | 74 | 73 | 84 | 51 | 81 |
| Intern-S1 | 63 | 67 | 64 | 41 | 64 |
| GLM-4.5V | 42 | 35 | 44 | 30 | 49 |
| Qwen2.5-VL-72B-Inst. | 37 | 32 | 48 | 23 | 50 |

## 5.3 ANALYSIS OF STEP-LEVEL ERROR TYPES IN MLLMs' PHYSICS REASONING

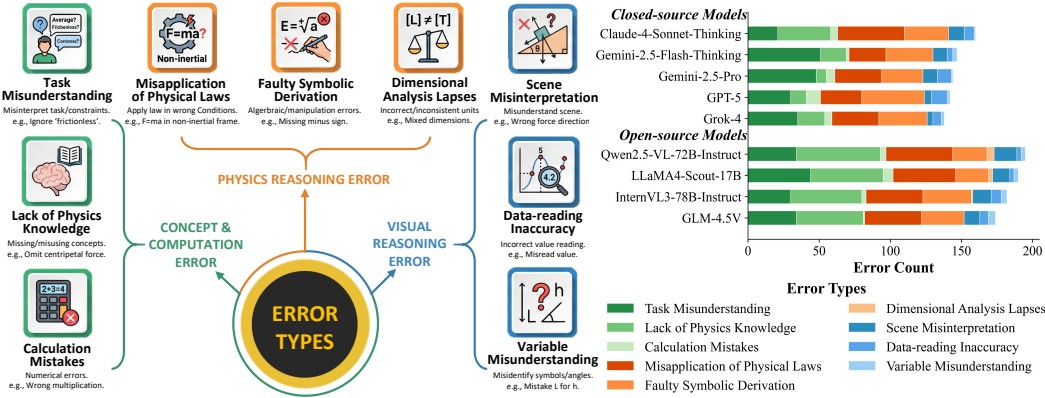

Figure 7: Statistical analysis of step-level error types in MLLMs' physics reasoning.

We perform a fine-grained error analysis on 85 problems from the HiPhO where both Gemini-2.5-Pro and GPT-5 scored $\leq 75\%$ of the total points. For each step-level criterion marked incorrect, we assign one of nine error subtypes under three main categories: (1) **Concept & Computation Error**, (2) **Physics Reasoning Error**, and (3) **Visual Reasoning Error**, as shown in Fig. 7. Our findings show that: (1) **Open-source models make notably more errors**, especially in physics knowledge and law application, indicating substantial room for improvement in domain understanding. (2) **Closed-source models mainly struggle with task misinterpretation and symbolic derivation**, underscoring the complexity of Olympiad-style multi-step reasoning. (3) **Scene misinterpretation is the most frequent visual error for open-source models**, highlighting persistent challenges with interpreting physical setup in figures. These patterns expose key weaknesses in both physics and visual reasoning. Representative examples of each error type are provided in Appendix F.

### 5.4 Analysis of Grading Strategies and Evaluator Consistency

**The Need for Fine-Grained Evaluation.** Relying only on answer-level scores can underestimate model performance, as incorrect answer extraction or mismatched yet equivalent final expressions may obscure otherwise valid intermediate steps. By taking max(answer-level, step-level), we align with human grading logic: correct final answers receive full credit, while incorrect ones still earn partial credit through valid steps, ensuring fairer evaluation. Further analysis is in Appendix E.1.

**LLM Graders vs. Human Experts.** We conduct a systematic comparison across four graders: **Gemini-2.5-Flash** (used in our work), **DeepSeek-R1** (open-source), **GPT-4o** (commonly used), and **human experts**. As shown in Table 13, Gemini-2.5-Flash consistently produced scores that closely matched human scores (typically within 1 point). DeepSeek-R1 also performed well and may serve as an open-source alternative. In contrast, GPT-4o shows significant deviation, often underestimating model performance. Further statistical analysis is provided in Appendix E.3.

### 5.5 How Can (M)LLMs Achieve True Human-Level Physics Reasoning?

**Model Limitations.** (M)LLMs face three structural limitations compared to human contestants. **(1) Multimodality**: LLMs cannot interpret diagrams, and MLLMs still struggle with variable-based or data-intensive figures. **(2) Generation**: Olympiads often require sketching functional plots, but most MLLMs cannot produce diagrams directly, and evaluating generated plots remains challenging. **(3) Embodiment**: IPhO, APhO, and EuPhO include separate experimental exams, yet models lack physical interaction capacity, so such problems are excluded. Achieving true human-level reasoning will therefore require advances in multimodality, generation, and embodiment, enabling models to integrate text and vision, generate diagrams, and engage in experimental reasoning.

**Experimental Physics.** To bridge the experimental gap, we envision three progressive directions. **(1) Multimodal experimental reasoning** can be tested using visual inputs of experimental setups and data. **(2) Simulated experimental environments** allow models to manipulate virtual apparatus and engage in interactive reasoning. **(3) Embodied experimentation** requires robotic agents to perform real-world operations, aligning with emerging Vision-Language-Action (VLA) paradigms.

## 6 Conclusion and Future Work

We presented HiPhO, the first benchmark to systematically evaluate (M)LLMs against top-performing students in 2024-2025 physics Olympiads. Uniquely, HiPhO integrates answer-level and step-level grading from official marking schemes, enabling rigorous, human-aligned evaluation. Comparing model scores with human contestants reveals clear performance gaps: SOTA closed-source reasoning MLLMs achieve gold-level performance on most exams but still trail the very best humans; most open-source MLLMs remain at bronze or below, while open-source LLMs obtain multiple golds. Our detailed analyses across modalities, physics fields, and error types further pinpoint persistent weaknesses in text with data figures, in optics, and in physical reasoning accuracy, underscoring the considerable gap that remains before reaching human-level mastery.

Looking ahead, HiPhO can evolve in three directions. First, expanding to include Olympiads such as CPhO and USAPhO would create a broader benchmark. Second, in the longer term, incorporating evaluations of experimental physics would allow a more authentic comparison with human reasoning, though this remains highly challenging. Third, building a reliable open-source grader would enable more accurate evaluation of scientific reasoning. By incorporating additional Olympiads, experimental tasks, and an open-source grader, future versions of HiPhO can provide a more comprehensive and human-aligned standard to advance multimodal physical reasoning.

## ETHICS STATEMENT

All authors have read and adhered to the ICLR Code of Ethics. This study does not involve sensitive personal information or applications with potential risks. All exams and human scores used in our benchmark are publicly available from official websites, with no proprietary or confidential data included. The authors declare that there are no potential conflicts of interest related to this work.

## REPRODUCIBILITY STATEMENT

To facilitate reproducibility, we offer an anonymous codebase at `https://anonymous.4open.science/r/HiPhO`, also provided in the supplementary material. The repository includes the proposed benchmark and evaluation pipeline for the physics Olympiads, together with a detailed README containing installation and usage instructions.

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

# Supplemental Material of HIPHO Benchmark

## A  THE USE OF LARGE LANGUAGE MODELS

In this work, GPT-5 was used solely as a general-purpose tool to polish the writing. For evaluation, we employed a combination of rule-based and model-based approaches, with Gemini-2.5-Flash serving as the grader. Fig. 7 was designed by the authors, with icons generated using Google Nano Banana. No other substantive use of LLMs was involved in this paper.

## B  DETAILS OF HIPHO BENCHMARK

### B.1  DATA SOURCE AND EXAM FORMAT

- **IPhO (International Physics Olympiad):** The most prestigious global physics Olympiad for high school students since 1967, featuring challenging theoretical and experimental exams.

- **APhO (Asian Physics Olympiad):** A regional contest launched in 2000 for Asian and Oceanian students, structured similarly to IPhO with both theory and laboratory components.

- **EuPhO (European Physics Olympiad):** Established in 2017, a multi-day competition for European students that emphasizes creative problem solving in both theory and experiment.

- **NBPhO (Nordic-Baltic Physics Olympiad):** A regional contest among Nordic and Baltic countries, primarily theoretical but also including experimental problems.

- **PanPhO (Pan Pearl River Delta Physics Olympiad):** An invitational competition for top schools from the Pearl River Delta and neighboring regions in China.

- **PanMechanics (Pan Pearl River Delta Mechanics Test):** A specialized subset of PanPhO focusing solely on mechanics, typically offered as a shorter, single-field exam.

- **F=MA:** A U.S. mechanics contest organized by the American Association of Physics Teachers, serving as the entry test for the U.S. Physics Olympiad (USAPhO).

Table 5: Source links for the 13 collected physics Olympiad exams in the HIPHO benchmark.

| Year | | 2025 | | | 2024 | |
|---|---|---|---|---|---|---|
| Physics Olympiad | Problem | Solution | Results | Problem | Solution | Results |
| IPhO | Q1, Q2, Q3 | S1, S2, S3 | results | Q1, Q2, Q3 | S1, S2, S3 | results |
| APhO | problem | solution | results | Human results are unavailable | | |
| EuPhO | problem | solution | results | problem | solution | results |
| NBPhO | problem | solution | results | problem | solution | results |
| PanPhO | Q1, Q2 | S1, S2 | results | Q1, Q2 | S1, S2 | results |
| PanMechanics | problem | solution | results | problem | solution | results |
| F=MA | problem | solution | results | problem | solution | results |

Physics Olympiads feature diverse exam formats (Table 6): IPhO, APhO, and EuPhO include a separate 20-point experimental exam, and NBPhO integrates theory and experiment in a single 72-point paper. Since (M)LLMs lack embodied experimental and diagram-drawing capabilities, we exclude such questions for fair evaluation, so the **Full Mark (Model)** in Table 3 reflects only the theoretical score. In addition, we introduce step-level scoring based on official marking schemes, which are available for 7 of 13 exams, with 4 supporting multiple solutions, ensuring greater fairness.

Table 6: Exam formats and scoring components of physics Olympiads in the HIPHO benchmark.

| Physics Olympiad | IPhO | | APhO | EuPhO | | NBPhO | | PanPhO | | PanMechanics | | F=MA | |
|---|---|---|---|---|---|---|---|---|---|---|---|---|---|
| Year | 2025 | 2024 | 2025 | 2025 | 2024 | 2025 | 2024 | 2025 | 2024 | 2025 | 2024 | 2025 | 2024 |
| Theoretical Exam Score | 30 | 30 | 30 | 30 | 30 | 72 | 72 | 100 | 100 | 100 | 100 | 25 | 25 |
| - **Theoretical Score** | 29.4 | 29.3 | 30 | 29 | 28 | 43.5 | 50 | 100 | 98 | 100 | 100 | 25 | 25 |
| - Diagram Score | 0.6 | 0.7 | – | 1 | 2 | 4.5 | – | – | 2 | – | – | – | – |
| - Experimental Score | – | – | – | – | – | 24 | 22 | – | – | – | – | – | – |
| Experimental Exam Score | 20 | 20 | 20 | 20 | 20 | – | – | – | – | – | – | – | – |
| Official Marking Scheme | ✓ | ✓ | ✓ | ✓ | ✓ | ✓ | ✓ | ✗ | ✗ | ✗ | ✗ | ✗ | ✗ |
| Multiple Solutions | ✗ | ✗ | ✓ | ✗ | ✓ | ✓ | ✓ | ✗ | ✗ | ✗ | ✗ | ✗ | ✗ |

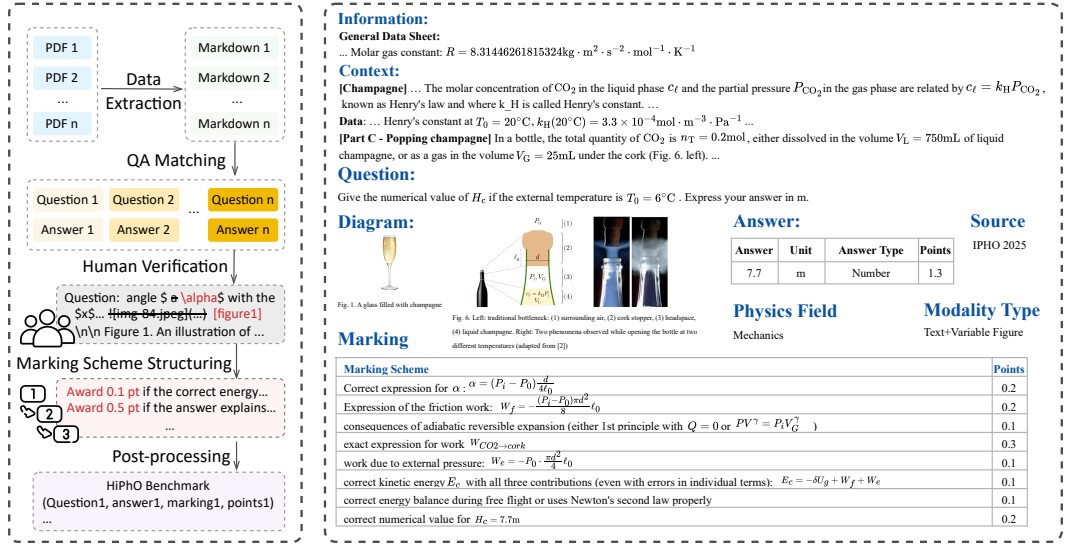

Figure 8: High-quality data processing pipeline (left) and an example of HIPHO data (right).

## B.2 DATA PROCESSING PIPELINE

We implemented a structured, multi-stage data processing pipeline, as illustrated in Fig. 8 (left).

- **Data Extraction.** Official Olympiad exam papers in PDF format were processed with OCR-based tools. Text blocks were reflowed into markdown for easier parsing and editing.

- **QA Matching.** Questions were aligned with their official answers by matching indices and numbering patterns, ensuring that every problem statement had a corresponding answer entry.

- **Human Verification.** Human experts carefully reviewed all extracted content. This included correcting OCR misrecognitions and ensuring textual consistency with the original exam.

- **Marking Scheme Structuring.** For exams with marking schemes, step-level scoring criteria were systematically extracted, and then reformatted into standardized, model-readable rules.

- **Post-Processing.** Verified QAs underwent additional refinement:

  - *Context Completion:* Long problem stems and necessary background information were consolidated to provide self-contained contexts.

  - *Subquestion Structuring:* Interdependent subparts were explicitly separated and re-labeled to reflect the intended order of solution.

  - *Unit Specification:* Required physical units were clarified in the problem statement to minimize ambiguity and avoid unfair penalization.

As shown in Fig. 8 (right), the finalized dataset is organized into a unified `json` format:

- **Information**: Physics constants table on the exam cover page (if available), e.g., $g = 9.8\,m/s^2$.

- **Context**: Problem stem content, including introductory descriptions or preceding subparts.

- **Question**: Specific question text, with multiple subquestions explicitly split into (1), (2), etc.

- **Diagram**: Path to required figures or diagrams associated with the question.

- **Marking**: Step-level scoring points in the unified form "Award xx pt if the answer . . .".

- **Answer**: Ground-truth solution including value, units, answer type, and allocated points.

- **Physics Field**: The physics domain of the problem (Mechanics, Electromagnetism, Thermodynamics, Optics, Modern Physics).

- **Modality Type**: The modality type of the problem (Text-Only, Text+Illustration Figure, Text+Variable Figure, Text+Data Figure).

- **Source**: The corresponding Olympiad exam, e.g., IPhO 2025.

### B.3 DATA LEAKAGE ANALYSIS

To supplement our main results in Section 3.4, we provide in Table 7 a summary of the data leakage check on the HIPHO subset (90 randomly sampled problems), with detailed breakdowns across all 13 Olympiads in Tables 8 and 9. Following the partial-prompt completion strategy (Wu et al., 2025), we prompted models with the first 60% of each problem and recorded their greedy continuations.

Table 8 reports the ROUGE-L (Lin, 2004) overlap between model continuations and ground-truth endings, showing that most models, closed-source and open-source alike, remain below a 20% completion rate, indicating minimal verbatim reproduction. Table 9 further shows that partial-prompt answer accuracy is nearly always 0%. These findings confirm that the HIPHO benchmark exhibits minimal data leakage and remains highly challenging.

Table 7: Summary of data leakage check (%) on the HIPHO subset.

| Closed-source Models | ROUGE-L | ACC | Open-source Models | ROUGE-L | ACC |
|---|---|---|---|---|---|
| Gemini-2.5-Pro | 11.8 | 1.7 | Intern-S1 | 10.9 | 2.2 |
| Gemini-2.5-Flash-Thinking | 11.8 | 1.7 | InternVL3-78B-Instruct | 12.2 | 1.1 |
| GPT-5 | 12.1 | 3.7 | Qwen2.5-VL-72B-Instruct | 10.8 | 1.1 |
| Grok-4 | 14.9 | 1.7 | DeepSeek-R1 | 10.3 | 0.0 |
| Claude-4-Sonnet-Thinking | 10.7 | 1.1 | Qwen3-235B-A22B | 8.4 | 1.1 |

Table 8: Partial-prompt completion rate (%) of 10 top-performing models across all 13 Olympiads.

| ROUGE-L | IPhO 2025 | IPhO 2024 | APhO 2025 | EuPhO 2025 | EuPhO 2024 | NBPhO 2025 | NBPhO 2024 | PanPhO 2025 | PanPhO 2024 | PanMechanics 2025 | PanMechanics 2024 | F=MA 2025 | F=MA 2024 |
|---|---|---|---|---|---|---|---|---|---|---|---|---|---|
| *Closed-source Models* | | | | | | | | | | | | | |
| Gemini-2.5-Pro | 15.9 | 15.2 | 14.9 | 6.5 | 14.3 | 17.2 | 15.9 | 13.3 | 13.9 | 4.1 | 5.1 | 6.1 | 4.9 |
| Gemini-2.5-Flash-Thinking | 11.5 | 11.4 | 10.0 | 13.5 | 18.5 | 7.2 | 7.6 | 12.6 | 11.2 | 6.9 | 7.4 | 16.2 | 19.3 |
| GPT-5 | 11.2 | 8.0 | 13.3 | 11.4 | 11.1 | 15.1 | 12.0 | 13.8 | 11.9 | 5.6 | 8.6 | 15.7 | 20.2 |
| Grok-4 | 12.2 | 10.9 | 13.4 | 15.0 | 22.1 | 15.7 | 14.9 | 16.0 | 13.1 | 5.1 | 10.3 | 18.2 | 29.3 |
| Claude-4-Sonnet-Thinking | 11.5 | 14.7 | 12.9 | 10.5 | 7.1 | 12.3 | 11.5 | 13.5 | 13.6 | 3.7 | 5.9 | 8.4 | 7.3 |
| *Open-source Models* | | | | | | | | | | | | | |
| Intern-S1 | 12.9 | 13.6 | 13.4 | 9.1 | 12.7 | 11.0 | 13.7 | 13.4 | 12.5 | 2.5 | 3.5 | 7.9 | 10.0 |
| InternVL3-78B-Instruct | 11.9 | 14.0 | 19.1 | 6.9 | 12.3 | 12.5 | 10.6 | 13.2 | 14.0 | 14.7 | 10.6 | 7.6 | 6.4 |
| Qwen2.5-VL-72B-Instruct | 12.5 | 12.4 | 18.2 | 10.3 | 11.1 | 11.2 | 10.3 | 12.4 | 12.2 | 3.0 | 7.4 | 4.6 | 7.5 |
| DeepSeek-R1 | 13.0 | 13.3 | 12.0 | 6.6 | 12.5 | 13.1 | 10.5 | 10.2 | 13.1 | 1.3 | 5.7 | 8.2 | 9.8 |
| Qwen3-235B-A22B | 12.0 | 13.8 | 10.9 | 7.9 | 9.7 | 8.0 | 8.3 | 7.7 | 11.5 | 0.0 | 1.5 | 6.5 | 5.8 |

Table 9: Partial-prompt Answer Accuracy (%) of 10 top-performing models across all 13 Olympiads.

| ACC | IPhO 2025 | IPhO 2024 | APhO 2025 | EuPhO 2025 | EuPhO 2024 | NBPhO 2025 | NBPhO 2024 | PanPhO 2025 | PanPhO 2024 | PanMechanics 2025 | PanMechanics 2024 | F=MA 2025 | F=MA 2024 |
|---|---|---|---|---|---|---|---|---|---|---|---|---|---|
| *Closed-source Models* | | | | | | | | | | | | | |
| Gemini-2.5-Pro | 0.0 | 0.0 | 0.0 | 0.0 | 8.3 | 0.0 | 0.0 | 9.1 | 0.0 | 0.0 | 0.0 | 0.0 | 0.0 |
| Gemini-2.5-Flash-Thinking | 0.0 | 0.0 | 0.0 | 0.0 | 8.3 | 0.0 | 0.0 | 9.1 | 0.0 | 0.0 | 0.0 | 0.0 | 0.0 |
| GPT-5 | 11.1 | 0.0 | 12.5 | 0.0 | 0.0 | 0.0 | 0.0 | 12.1 | 0.0 | 0.0 | 0.0 | 0.0 | 0.0 |
| Grok-4 | 11.1 | 0.0 | 0.0 | 0.0 | 8.3 | 0.0 | 0.0 | 0.0 | 0.0 | 0.0 | 0.0 | 0.0 | 0.0 |
| Claude-4-Sonnet-Thinking | 0.0 | 0.0 | 0.0 | 0.0 | 0.0 | 0.0 | 0.0 | 0.0 | 0.0 | 0.0 | 14.3 | 0.0 | 0.0 |
| *Open-source Models* | | | | | | | | | | | | | |
| Intern-S1 | 11.1 | 0.0 | 0.0 | 0.0 | 0.0 | 0.0 | 0.0 | 0.0 | 0.0 | 0.0 | 14.3 | 0.0 | 0.0 |
| InternVL3-78B-Instruct | 0.0 | 0.0 | 0.0 | 0.0 | 0.0 | 0.0 | 0.0 | 0.0 | 0.0 | 0.0 | 0.0 | 0.0 | 16.7 |
| Qwen2.5-VL-72B-Instruct | 0.0 | 0.0 | 0.0 | 0.0 | 0.0 | 0.0 | 0.0 | 0.0 | 0.0 | 0.0 | 14.3 | 0.0 | 0.0 |
| DeepSeek-R1 | 0.0 | 0.0 | 0.0 | 0.0 | 0.0 | 0.0 | 0.0 | 0.0 | 0.0 | 0.0 | 0.0 | 0.0 | 0.0 |
| Qwen3-235B-A22B | 11.1 | 0.0 | 0.0 | 0.0 | 0.0 | 0.0 | 0.0 | 0.0 | 0.0 | 0.0 | 0.0 | 0.0 | 0.0 |

## C  DETAILS OF EVALUATION FRAMEWORK

### C.1  INFERENCE PROMPT

To ensure consistent and reproducible evaluation across models, we adopt a compact instruction template that: (i) enforces LaTeX-formatted math, (ii) separates full reasoning from the final answer using <think> and <answer> tags, and (iii) standardizes multi-part and multiple-choice outputs with boxed answers.

To align with the original language of each Olympiad exam, we use English prompts for English-language exams and Chinese prompts for Chinese-language exams. Specifically, the language settings are as follows:

- **English Exams:** IPhO, APhO, EuPhO, NBPhO, PanPhO, F=MA

- **Chinese Exam:** PanMechanics

For completeness, we provide both the English and Chinese prompt templates below.

---

**Inference Prompt for English Exams**

You are participating in a high school physics Olympiad exam.

Please read the following question carefully and provide a clear, step-by-step solution with full reasoning.

Instructions:
1. Use LaTeX to format all variables, equations, and calculations.
2. Enclose your full reasoning process within <think></think> tags.
3. Provide the final answer within <answer></answer> tags, using the format of [\boxed{answer}]. Do not include units inside the box.
4. For multiple sub-questions, list the answers in order using the format: [\boxed{answer1}, \boxed{answer2}, ...].
5. For multiple-choice questions, provide the final selected option(s) in the boxed answer instead of the calculation result (e.g., [\boxed{A}]).

Example of Output:
<think>
Step 1: Analyze the problem... Step 2: Apply the relevant equations...
</think>
<answer>
[\boxed{A}, \boxed{3.2}]
</answer>

Useful information (formulas, constants, units, if applicable):
{information}

Context (if applicable):
{context}
Question (Answer only the question stated below):
{question}

---

---

**Inference Prompt for Chinese Exams**

你正在参加高中物理竞赛。

请仔细阅读下列题目，结合上下文信息，详细推导并给出清晰、有条理的解题步骤与完整的逻辑推理过程。

作答要求：
1. 所有物理量、公式和计算过程须使用LaTeX 格式书写。
2. 将完整的推理过程用<think>和</think>标签括起来。
3. 将最终答案置于<answer>和</answer>标签中，答案格式为[\boxed{答案}]，方框内不包含单位。
4. 对于包含多个小问的题目，按顺序列出所有答案，格式为：[\boxed{答案1}, \boxed{答案2}, ...]。
5. 对于选择题，请在答案的方框中给出最终选择的选项，而不是计算结果（例如：[\boxed{A}]）。

输出示例：
<think>
第一步：分析问题... 第二步：运用相关公式...
</think>
<answer>
[\boxed{A}, \boxed{3.2}]
</answer>

可用信息（如物理公式、常数、单位等）：
{information}

背景信息（如有）：
{context}

题目内容（仅回答以下问题）：
{question}

---

## C.2 ANSWER-LEVEL COARSE-GRAINED SCORE

The answer-level coarse-grained evaluation pipeline extends the PHYSICS framework (Zheng et al., 2025) and follows a structured sequence to assess whether the model's final answers are correct. The process comprises the following key steps:

- **Answer Extraction.** Final answers are extracted from the model's solution using automatic parsing of boxed expressions (e.g., \boxed{...}). This ensures that evaluation targets the model's intended outputs, while ignoring intermediate reasoning steps.

- **Rule-Based Matching.** A rule-based math-verifier (Kydlíček) compares the extracted answers with the ground-truth answers. It performs correct matching on either numeric or symbolic expressions, while also accounting for units and answer types.

- **Model-Based Verification.** If no exact match is found, a second-pass verification is performed using a powerful judge model (Gemini-2.5-Flash). This model compares the model's answer and the ground-truth answer to determine physical or mathematical equivalence, thereby reducing false negatives caused by alternative expressions, equivalent transformations, or formatting inconsistencies.

- **Multi-Part Matching Logic.** For problems consisting of multiple sub-questions (e.g., labeled as (1), (2), etc.), the intended answer order is explicitly defined by the problem statement. To reflect this, we extract boxed answers from the model's response in reverse order and compare them sequentially with the ground-truth, following the original question structure.

Unlike the original PHYSICS's evaluation framework, which relied on a fine-tuned lightweight 8B verifier, our pipeline leverages Gemini-2.5-Flash for model-based verification. This substitution sub-

stantially improves the reliability of equivalence judgments. The corresponding evaluation prompt is adapted from the PHYSICS design (Zheng et al., 2025), as shown below.

---

**Prompt for Answer-level Model-based Evaluation**

You are a diligent and precise assistant tasked with evaluating the correctness of responses. You will receive a question, an output sentence, and the correct answer. Your task is to determine if the output sentence accurately answers the question based on the provided correct answer. Respond with either [Correct] or [Incorrect].

Special considerations:
1. **Multiple Answers:** If the output contains multiple answers, evaluate whether later answers modify or correct earlier ones. In such cases, compare the final answer with the correct answer. If the final answer is unclear or incorrect, respond with [Incorrect].
2. **Mathematical Problems:** If the formats differ but the answers are mathematically equivalent such as 256/55=4.65, respond with [Correct].
3. **Physics Problems:** If the values match such as 3=3 GHz, respond with [Correct].
4. **Explicit Options:** If the question provides explicit candidate answers, the output will be considered correct if it clearly indicates the correct option's code or the correct option's content.
5. **No Explicit Options:** If the question does not provide explicit options, the output must align with the correct answer in content and meaning to be considered [Correct].

Question: {problem}
Output sentence: {given_answer}
Correct answer: {ground_truth}

Final Instruction:
You must respond with exactly one of the following: [Correct] or [Incorrect].
Do NOT include any explanation, reasoning, or additional text.
Any deviation from this format (even a single word) will be considered INVALID.

Judgement:

---

### C.3   STEP-LEVEL FINE-GRAINED SCORE

The step-level fine-grained evaluation measures the model's reasoning quality by comparing its solution steps against detailed criteria from official marking schemes, with each criterion representing a specific conceptual, physical, or mathematical step. The evaluation follows the steps below:

- **Parse Marking Criteria.** Each problem is accompanied by a list of step-level marking points, such as "Applies energy conservation correctly" or "Derives the correct force expression." These marking points are parsed from the dataset and include descriptions and assigned partial scores.

- **Model-Based Step Scoring.** For each marking point, the model's solution is independently assessed using Gemini-2.5-Flash, a strong judge model. A dedicated prompt is constructed to instruct the model to evaluate whether the student's solution satisfies the given criterion and to return a numerical score (e.g., 1.0, 0.5, or 0.0) accordingly. This approach enables semantic and physical equivalence checking beyond surface-level matching.

- **Score Aggregation.** The scores assigned for each marking point are aggregated to compute the final fine-grained score for the question. These scores are stored alongside their corresponding criteria for detailed feedback and analysis. For EuPhO and NBPhO problems with multiple official marking schemes, we take the highest score across schemes, reflecting the fact that solutions may follow different valid approaches.

Compared to commonly used answer-level evaluation, this fully model-based step-level evaluation provides a more accurate and context-aware assessment of the model's reasoning. The fine-grained scoring closely aligns with how human graders assign partial credit in real Olympiad exams, mak-

ing it one of the core innovations of our benchmark and a more faithful reflection of true exam performance. The prompt used for step-level model-based evaluation is shown below.

---

**Prompt for Step-level Model-based Evaluation**

You are an expert physics competition grader. Evaluate the student's solution against the specific grading criterion.

Physics Problem:
{question}

Student's Solution:
{prediction}

Grading Criterion:
{criterion['description']}

Instructions:
1. Analyze the student's solution for physics concepts, mathematical derivations, and calculations.
2. Award points strictly according to the criterion.
3. Consider both conceptual understanding and technical accuracy.

Critical:
1. You MUST respond with ONLY a single number (e.g., 1.0, 0.5, 0.0).
2. NO explanations, NO text, NO reasoning - JUST THE NUMBER.
3. If you provide any text other than the number, your response will be invalid.

Score:

---

## D    DETAILS OF EXPERIMENTS AND RESULTS

### D.1    EXPERIMENTAL SETUP

**Experimental Setup for LLMs vs. MLLMs.**    In our experiments, MLLMs receive both the problem text and any accompanying figures as input, while LLMs are provided with text only. This setting reflects real exam conditions, where LLMs can process textual information but cannot directly interpret images. Extracting visual information for LLMs would require an additional MLLM as a pre-processor, and the choice of extraction model would inevitably affect the results. More importantly, such a setup would no longer represent the LLM's independent problem-solving ability, since external assistance is involved. Therefore, in our evaluation, LLMs are tested solely on textual input. Although this may lower their scores on multimodal questions, it provides a fairer measure of their standalone reasoning and imaginative capabilities.

**Hyperparameter Settings.** We adopt VLMEvalKit (Duan et al., 2024) as the evaluation framework for benchmarking MLLMs. To reduce randomness and improve the reliability of evaluation, each problem is tested using **eight independent inference runs** at a **temperature of 0.6**; its score is averaged across runs, and the exam score is the sum of these averages. This setup helps reduce score fluctuations caused by randomness in the model's responses. In addition, the **maximum token limit** is set in reference to the largest value permitted by each model, helping prevent output truncation and ensuring that responses are complete and valid.

**Evaluated Models.** To assess the physics reasoning capabilities of state-of-the-art (M)LLMs relative to top-performing human contestants, we evaluate 30 representative models, including 11 closed-source MLLMs, 11 open-source MLLMs, and 8 open-source LLMs, based on the following criteria: **(1) Recency:** Most models were released after April 2025, with the newest launched in August 2025 (e.g., GPT-5). **(2) Diversity:** The selection includes both closed- and open-source models, covering reasoning-specialized and general-purpose architectures from a wide range of de-

velopers. **(3) Model Scale:** A range of model sizes—small, medium, and large—is included to enable performance comparisons by scale. Specifically,

**11 Closed-source MLLMs:** GPT-5 (OpenAI, b), o3 (OpenAI, c), o4-mini (high) (OpenAI, c), o4-mini (OpenAI, c), GPT-4o (OpenAI, a), Gemini-2.5 Series (Comanici et al., 2025), Grok-4 (xAI), Claude-4-Sonnet(-Thinking) (Anthropic), Mistral-Medium-3 (Mistral).

**11 Open-source MLLMs:** Intern-S1 (Bai et al., 2025a), InternVL3 Series (Zhu et al., 2025), Qwen2.5-VL Series (Bai et al., 2025b), GLM-4.5V (Team et al., 2025b), DeepSeek-VL2 (Wu et al., 2024), LLaMA4-Scout-17B (Meta), Phi-4-multimodal (Abouelenin et al., 2025).

**8 Open-source LLMs:** GPT-OSS (Agarwal et al., 2025), Kimi-K2-Instruct (Team et al., 2025a), DeepSeek-R1 (Guo et al., 2025), DeepSeek-V3 (Liu et al., 2024), Qwen3 Series (Yang et al., 2025).

### D.2 Illustration of Medal Scorelines

- **IPhO, APhO, EuPhO:** These include separate theoretical and experimental exams, with medals awarded on total scores. As (M)LLMs cannot do experiments, we use the lowest theoretical exam score of gold medalists as the gold cutoff, and analogously for silver and bronze.

- **NBPhO:** Theory and experiment appear in the same paper, with medals based on total scores. We set the gold line as the lowest total score of gold medalists. Since the paper contains experimental and plotting items, the model's full mark is lower than the human's; such items are counted as zero to reflect current limitations of (M)LLMs.

- **PanPhO, PanMechanics:** These are theory-only exams, but the official website reports scores only for Hong Kong SAR contestants. Thus, thresholds are based on the lowest scores of Hong Kong SAR medalists. If scores of all medalists were available, the top-1 human score would likely be higher, while the medal thresholds could be lower.

- **F=MA:** With over 5,000 participants, only score histograms are published and no official medals are awarded. As a USAPhO qualifier, the cutoff score for advancement is treated as the gold line. Silver and bronze thresholds are inferred from Physics Bowl conventions (top 20% and 35%), estimated from the histogram, while the gold line also aligns with the Physics Bowl top-10% rule.

### D.3 Leaderboard Analysis

To better illustrate the overall ranking of all models, we provide a comprehensive leaderboard in Table 10. Based on this view, several observations can be made:

- **Dominance of top closed-source models.** The top five positions are all occupied by leading closed-source models, each winning 10–12 gold medals. This clear gap demonstrates the exceptional multimodal physics reasoning capability of closed-source MLLMs.

- **Progress of open-source models.** Open-source models are steadily closing the gap. Notably, DeepSeek-R1 surpasses the closed-source Claude-4-Sonnet-Thinking on the leaderboard, despite lacking multimodal input. Similarly, Qwen3-235B-A22B and Qwen3-32B both outperform o4-mini and o4-mini (high), highlighting strong reasoning ability.

- **Gap between open-source MLLMs and LLMs.** Overall, open-source MLLMs remain weaker than open-source LLMs. While enhancing multimodal understanding, they still need to strengthen physics reasoning and maintain textual reasoning ability, which also points to future directions for open-source MLLM development.

- **Medal distribution across Olympiads.** The overall medal distribution reveals a clear pattern: from left to right, exam difficulty decreases and gold medals increase, which aligns with our difficulty categorization.

## E Statistical Analysis of Evaluation Metrics and Grading Models

### E.1 Sensitivity of Different Evaluation Metrics

We conducted a sensitivity analysis using eight independent grading runs per model. In addition to reporting the answer-level and step-level scores, we also include their average and maximum. As

Table 10: Leaderboard of the HIPHO benchmark (13 physics Olympiads from 2024–2025) using the *exam score* metric. Gold, Silver and Bronze indicate scores above the respective thresholds. Models are ranked by medal counts; **bold** is the highest score, and underline is the second highest. Here, only the theoretical parts of exams are used, hence Full Mark (Model) $\leq$ Full Mark (Human).

| Physics Olympiad Year | IPhO 2025 | IPhO 2024 | APhO 2025 | EuPhO 2025 | EuPhO 2024 | NBPhO 2025 | NBPhO 2024 | PanPhO 2025 | PanPhO 2024 | PanMechanics 2025 | PanMechanics 2024 | F=MA 2025 | F=MA 2024 | Medal Table | | |
|---|---|---|---|---|---|---|---|---|---|---|---|---|---|---|---|---|
| Full Mark (Human) | 30.0 | 30.0 | 30.0 | 30.0 | 30.0 | 72.0 | 72.0 | 100.0 | 100.0 | 100.0 | 100.0 | 25.0 | 25.0 | | | |
| Full Mark (Model) | 29.4 | 29.3 | 30.0 | 29.0 | 28.0 | 43.5 | 50.0 | 100.0 | 98.0 | 100.0 | 100.0 | 25.0 | 25.0 | | | |
| Top-1 Score (Human) | 29.2 | 29.4 | 30.0 | 27.0 | 30.0 | 53.2 | 40.8 | 81.0 | 66.5 | 62.0 | 51.0 | 25.0 | 24.0 | | | |
| Top-1 Score (Model) | 22.7 | 25.9 | 27.9 | 14.9 | 21.9 | 34.1 | 35.9 | 60.3 | 75.4 | 72.1 | 79.0 | 22.8 | 22.4 | | | |
| Gold Medal | 19.7 | 20.8 | 23.3 | 16.5 | 20.4 | 28.6 | 26.5 | 41.5 | 52.0 | 52.0 | 51.0 | 15.0 | 14.0 | | | |
| Silver Medal | 12.1 | 11.1 | 18.7 | 9.8 | 14.2 | 20.1 | 19.4 | 28.5 | 37.5 | 36.0 | 26.0 | 11.0 | 12.0 | | | |
| Bronze Medal | 7.2 | 3.6 | 13.1 | 5.8 | 8.9 | 15.2 | 13.5 | 14.5 | 16.0 | 20.0 | 12.0 | 9.0 | 10.0 | | | |
| Gemini-2.5-Pro | **22.7** | **25.9** | **27.9** | **14.9** | 21.8 | 32.3 | **35.9** | **60.3** | 64.1 | 69.5 | 70.2 | **22.8** | 22.0 | 12 | 1 | 0 |
| Gemini-2.5-Flash-Thinking | 20.2 | 23.9 | 27.4 | 13.2 | **21.9** | 29.0 | 29.3 | 44.6 | 54.9 | 60.5 | 55.9 | 17.8 | 19.1 | 12 | 1 | 0 |
| GPT-5 | 22.3 | 20.2 | 27.0 | 10.3 | 21.7 | 32.9 | 32.8 | 55.9 | 69.8 | 69.4 | **79.0** | 22.4 | **22.4** | 11 | 2 | 0 |
| o3 | 15.7 | 23.7 | 25.9 | 11.4 | 21.6 | **34.1** | 33.5 | 47.3 | 55.9 | 71.4 | 75.6 | 22.0 | 20.6 | 11 | 2 | 0 |
| Grok-4 | 18.7 | 23.5 | 25.0 | 11.5 | 20.5 | 25.8 | 29.3 | 45.0 | **75.4** | **72.1** | 78.6 | 19.8 | 19.8 | 10 | 3 | 0 |
| DeepSeek-R1 | 18.5 | 24.6 | 25.4 | 10.8 | 21.4 | 26.3 | 20.5 | 42.2 | 47.4 | 65.4 | 72.5 | 18.3 | 18.5 | 8 | 5 | 0 |
| Claude-4-Sonnet-Thinking | 19.0 | 22.0 | 24.8 | 9.7 | 20.5 | 28.1 | 25.6 | 43.1 | 39.3 | 57.4 | 61.8 | 19.2 | 20.1 | 8 | 4 | 1 |
| Qwen3-235B-A22B | 17.8 | 23.8 | 26.0 | 9.2 | 21.5 | 28.4 | 31.1 | 42.3 | 49.1 | 63.1 | 44.6 | 18.4 | 17.4 | 8 | 4 | 1 |
| Qwen3-32B | 15.7 | 19.3 | 23.9 | 9.8 | 21.2 | 28.9 | 24.1 | 36.6 | 41.8 | 67.0 | 59.2 | 18.9 | 16.6 | 7 | 6 | 0 |
| o4-mini | 15.4 | 22.9 | 22.8 | 10.1 | 20.9 | 26.9 | 27.3 | 39.4 | 47.1 | 64.2 | 62.5 | 18.6 | 18.5 | 7 | 6 | 0 |
| o4-mini (high) | 16.0 | 23.7 | 22.9 | 12.0 | 20.1 | 27.4 | 29.8 | 41.4 | 50.9 | 69.1 | 67.3 | 18.6 | 18.8 | 6 | 7 | 0 |
| Kimi-K2-Instruct | 16.5 | 19.8 | 24.2 | 11.0 | 16.9 | 26.5 | 26.2 | 35.9 | 41.8 | 65.9 | 58.9 | 16.0 | 18.2 | 5 | 8 | 0 |
| GPT-OSS-120B | 16.9 | 21.4 | 22.8 | 9.1 | 19.9 | 26.0 | 25.8 | 37.4 | 41.8 | 57.1 | 59.7 | 17.8 | 17.6 | 5 | 7 | 1 |
| Intern-S1 | 15.9 | 14.2 | 21.7 | 9.0 | 16.6 | 23.0 | 20.5 | 41.1 | 50.3 | 60.4 | 57.4 | 18.4 | 19.5 | 4 | 8 | 1 |
| Qwen3-30B-A3B | 13.6 | 15.4 | 22.7 | 9.8 | 16.5 | 24.7 | 21.7 | 31.9 | 39.5 | 49.9 | 45.0 | 15.5 | 15.0 | 2 | 11 | 0 |
| Claude-4-Sonnet | 15.7 | 19.2 | 22.8 | 9.5 | 16.5 | 27.5 | 21.3 | 40.4 | 43.3 | 46.5 | 48.5 | 16.8 | 16.5 | 2 | 10 | 1 |
| DeepSeek-V3 | 13.6 | 16.4 | 22.1 | 7.1 | 17.2 | 21.1 | 17.3 | 37.2 | 35.0 | 48.4 | 46.5 | 14.1 | 15.6 | 1 | 9 | 3 |
| Mistral-Medium-3 | 14.2 | 14.1 | 19.9 | 8.5 | 12.2 | 20.4 | 19.6 | 30.8 | 28.6 | 32.9 | 36.1 | 13.9 | 14.1 | 1 | 8 | 4 |
| GPT-4o | 10.2 | 9.4 | 15.1 | 6.8 | 9.2 | 16.4 | 11.7 | 27.8 | 22.8 | 28.2 | 26.5 | 15.0 | 10.9 | 1 | 1 | 10 |
| InternVL3-78B-Instruct | 12.9 | 12.5 | 17.7 | 7.5 | 15.2 | 22.3 | 22.5 | 26.2 | 27.4 | 21.1 | 27.1 | 12.0 | 13.0 | 0 | 8 | 5 |
| GLM-4.5V | 11.9 | 4.4 | 16.2 | 8.7 | 14.1 | 19.5 | 14.0 | 18.5 | 16.0 | 47.8 | 39.0 | 13.0 | 13.8 | 0 | 4 | 9 |
| Qwen3-8B | 10.6 | 12.7 | 11.5 | 7.1 | 19.8 | 20.1 | 17.3 | 26.3 | 22.3 | 21.8 | 22.8 | 10.8 | 10.0 | 0 | 2 | 10 |
| Qwen2.5-VL-72B-Instruct | 10.6 | 7.2 | 13.6 | 6.1 | 8.1 | 13.3 | 11.4 | 26.8 | 18.2 | 24.0 | 28.5 | 13.5 | 9.8 | 0 | 2 | 7 |
| LLaMA4-Scout-17B | 9.7 | 9.5 | 13.1 | 5.4 | 10.4 | 22.8 | 12.8 | 26.6 | 24.1 | 35.4 | 34.5 | 8.1 | 6.4 | 0 | 2 | 7 |
| Qwen2.5-VL-32B-Instruct | 9.9 | 8.2 | 16.5 | 6.9 | 10.0 | 15.3 | 14.4 | 22.5 | 22.4 | 28.1 | 29.9 | 7.6 | 4.6 | 0 | 1 | 10 |
| InternVL3-38B-Instruct | 8.9 | 7.8 | 12.3 | 6.1 | 8.3 | 14.0 | 10.6 | 24.1 | 20.4 | 27.5 | 24.8 | 8.2 | 6.8 | 0 | 0 | 7 |
| InternVL3-9B-Instruct | 4.7 | 3.7 | 7.2 | 4.2 | 4.1 | 9.4 | 6.0 | 12.4 | 11.0 | 11.6 | 16.3 | 6.4 | 6.2 | 0 | 0 | 2 |
| Qwen2.5-VL-7B-Instruct | 3.5 | 2.5 | 5.7 | 4.4 | 3.6 | 7.3 | 5.5 | 14.7 | 7.6 | 9.8 | 11.5 | 4.4 | 3.5 | 0 | 0 | 1 |
| Phi-4-multimodal | 2.0 | 1.6 | 4.2 | 3.6 | 3.6 | 5.0 | 4.5 | 8.3 | 9.0 | 10.0 | 10.1 | 4.4 | 5.0 | 0 | 0 | 0 |
| DeepSeek-VL2 | 1.8 | 0.5 | 2.5 | 3.4 | 3.4 | 5.0 | 3.2 | 5.6 | 4.8 | 7.3 | 6.4 | 5.0 | 3.9 | 0 | 0 | 0 |

shown in Table 11, all four scoring metrics exhibit strong internal consistency. The rankings based on maximum scores generally align with those based on answer- and step-level scores, indicating stable relative performance across metrics. Moreover, step-level scoring helps distinguish model performance more finely, as it awards partial credit for intermediate reasoning steps even when the final answer is incorrect.

We adopt the maximum of answer- and step-level scores to align with human grading logic: if the final answer is correct, full marks are awarded; if it is incorrect, partial credit is given based on intermediate steps. Therefore, we typically select the higher of the two scores as the final score per problem. This approach captures the benefits of step-level granularity while avoiding underestimation of correct final answers.

Table 11: Sensitivity analysis of different evaluation metrics.

| Olympiad Score | IPhO 2025 | | | | IPhO 2024 | | | |
|---|---|---|---|---|---|---|---|---|
| | answer-level | step-level | average | **max** | answer-level | step-level | average | **max** |
| Gemini-2.5-Pro | 18.2 | 20.1 | 19.1 | **22.7** | 24.4 | 25.7 | 25.1 | **25.9** |
| Gemini-2.5-Flash-Thinking | 16.1 | 19.2 | 17.6 | **20.2** | 20.1 | 23.7 | 21.9 | **23.9** |
| Claude-4-Sonnet-Thinking | 16.0 | 17.3 | 16.7 | **19.0** | 20.2 | 21.5 | 20.8 | **22.0** |

## E.2 IMPACT OF STEP-LEVEL SCORING

To quantify the impact of step-level grading, we analyzed how frequently step-level scores exceeded answer-level scores. As shown in Table 12, across three inference runs on IPhO 2025, step-level scores were the dominant contributor in 25.6% to 46.2% of the problems, depending on the model and run. This highlights the substantial contribution of step-level evaluation in fairly recognizing partial reasoning competence. Relying solely on answer-level scores can lead to underestimated performance, as incorrect answer extraction or mismatched yet equivalent final answers may obscure valid intermediate reasoning. More importantly, our scoring methodology is aligned with official rubrics, which assign partial points for intermediate steps. This alignment ensures that model scores can be directly and meaningfully compared to human contestants under the same criteria.

Table 12: Frequency of step-level score dominance on IPhO 2025.

| Model | Run 1 | Run 2 | Run 3 |
|---|---|---|---|
| Gemini-2.5-Pro | 30.8% | 30.8% | 25.6% |
| Gemini-2.5-Flash-Thinking | 33.3% | 41.0% | 46.2% |
| Claude-4-Sonnet-Thinking | 41.0% | 35.9% | 35.9% |

## E.3 IMPACT OF DIFFERENT GRADING MODELS

Many recent physics benchmarks rely on proprietary graders, e.g., GPT-4o (Xu et al., 2025; Feng et al., 2025; Dai et al., 2025). However, our empirical analysis shows that the widely used GPT-4o grader introduces noticeable discrepancies compared with human experts, motivating our choice of Gemini-2.5-Flash as a more reliable alternative for HIPHO.

To validate this choice, we conducted a systematic comparison across four graders: **Gemini-2.5-Flash** (used in our work), **DeepSeek-R1** (open-source), **GPT-4o** (commonly used), and **human experts**. For each of three evaluated models, we independently graded three inference runs using all four graders. As shown in Table 13, Gemini-2.5-Flash consistently produced scores that closely matched human scores (typically within 1 point). DeepSeek-R1 also performed well and may serve as an open-source alternative. In contrast, GPT-4o shows significant deviation, often underestimating model performance, especially on IPhO 2024.

Table 13: Comparison of different graders with human experts.

| Olympiad Grader | IPhO 2025 | | | | IPhO 2024 | | | |
|---|---|---|---|---|---|---|---|---|
| | Gemini-2.5-Flash | DeepSeek-R1 | GPT-4o | Human | Gemini-2.5-Flash | DeepSeek-R1 | GPT-4o | Human |
| Gemini-2.5-Pro | 22.1 | 21.9 | 21.5 | 22.0 | 25.7 | 24.0 | 21.1 | 26.0 |
| Gemini-2.5-Flash-Thinking | 19.9 | 20.0 | 20.4 | 19.1 | 23.5 | 22.5 | 20.0 | 23.3 |
| Claude-4-Sonnet-Thinking | 19.3 | 20.2 | 18.5 | 19.0 | 22.1 | 21.0 | 17.4 | 21.3 |

To further quantify grader reliability, we conducted a statistical analysis of score consistency using both mean absolute error and Pearson correlation against human expert scores. As shown in Table 14, Gemini-2.5-Flash consistently achieves the **highest correlations (0.88-0.95)** across both IPhO 2025 and IPhO 2024, along with the **smallest mean absolute errors (0.41-0.74)**, providing strong evidence of its alignment with expert judgment. In comparison, GPT-4o exhibits substantially lower correlation (0.41) and much larger error (2.41) on average.

Table 14: Statistical analysis of different graders.

| Metric Grader | Mean Absolute Error (↓) | | | Pearson Correlation (↑) | | |
|---|---|---|---|---|---|---|
| | Gemini-2.5-Flash | DeepSeek-R1 | GPT-4o | Gemini-2.5-Flash | DeepSeek-R1 | GPT-4o |
| IPhO 2025 | 0.41 | 0.88 | 0.79 | 0.95 | 0.90 | 0.75 |
| IPhO 2024 | 0.74 | 1.60 | 4.02 | 0.88 | 0.74 | 0.78 |
| **All (Avg)** | **0.58** | **1.24** | **2.41** | **0.95** | **0.86** | **0.41** |

This provides an important insight from our study: **the choice of grader plays a critical role in benchmark evaluation but is often overlooked**. Instead of relying on a commonly used grader, we carefully examined the reliability of different graders. Our future work will continue in this direction to develop an open-source, fine-tuned grader to further strengthen the benchmark.

# F  CASE STUDIES OF ERROR TYPES

To systematically analyze model reasoning failures, we define a hierarchical taxonomy of **nine fine-grained error types** grouped into **three major categories** (Fig. 7). This taxonomy is used to annotate step-level errors throughout our benchmark analysis.

- **Concept & Computation Error**
  These errors stem from missing conceptual understanding or simple numerical mistakes:

  - *Task Misunderstanding*: Misinterpretation of the problem setup or constraints. *E.g., ignoring the word "frictionless" or misunderstanding what is being asked.*

  - *Lack of Physics Knowledge*: Missing or misusing fundamental concepts. *E.g., failing to account for centripetal force in circular motion.*

  - *Calculation Mistakes*: Arithmetic or algebraic miscalculations. *E.g., errors in basic multiplication or sign mistakes.*

- **Physics Reasoning Error**
  These involve incorrect symbolic or physical reasoning:

  - *Misapplication of Physical Laws*: Applying laws in invalid contexts. *E.g., using $F = ma$ in a non-inertial frame without adding fictitious forces.*

  - *Faulty Symbolic Derivation*: Mistakes in symbolic manipulation. *E.g., dropping a minus sign or incorrectly simplifying an expression.*

  - *Dimensional Analysis Lapses*: Inconsistent or invalid unit reasoning. *E.g., mixing quantities of different dimensions, such as length and time.*

- **Visual Reasoning Error**
  These arise from misinterpreting diagrams or data:

  - *Scene Misinterpretation*: Misunderstanding visual layouts or physical setups. *E.g., drawing force vectors in the wrong direction.*

  - *Data-reading Inaccuracy*: Misreading numerical values from tables or plots. *E.g., interpreting a score of 5 as 4.2 in the data plot.*

  - *Variable Misunderstanding*: Confusing symbols or geometrical labels. *E.g., mixing up $L$ and $h$ in inclined plane problems.*

Each error type is labeled at the step level and contributes to our statistical and qualitative analysis in Section 5.3. More annotated examples are provided throughout this appendix for reference.

---

### *Case 1 Concept & Computation Error - Task Misunderstanding*

**Problem:**
**[Precession of the Earth's Axis]**
[Introduction]
It has long been known that the Earth's rotational axis undergoes precession, with ancient estimates around 29000 years and modern measurements around 25800 years. In this problem, you are asked to investigate this phenomenon using Newtonian mechanics.
You may use the following constants: $G = 6.67 \times 10^{-11}\,\mathrm{N\,m^2/kg^2}$, $R = 6.371 \times 10^6\,\mathrm{m}$, $M_E = 5.972 \times 10^{24}\,\mathrm{kg}$, $d_{SE} = 1.496 \times 10^{11}\,\mathrm{m}$, $M_S = 1.989 \times 10^{30}\,\mathrm{kg}$, $d_{ME} = 3.844 \times 10^8\,\mathrm{m}$, $M_M = 7.348 \times 10^{22}\,\mathrm{kg}$, $\alpha = 23.5°$.
[Part A: Shape of the Earth]
Because the Earth is not a perfect sphere, the Sun and Moon exert a nonzero torque. The Earth can be approximated as a uniformly dense fluid droplet forming an oblate spheroid. The small difference between equatorial and polar radii, $h_{\max} = R_e - R_p$, satisfies

$$h_{\max} \propto G^{-1}\omega^\beta M_E^\gamma R^\delta.$$

(A.1) Determine $\beta$, $\gamma$, and $\delta$. (A.2) Assuming the coefficient is 1, compute $h_{\max}$ in km. Use $h_{\max} = 21\,\mathrm{km}$ in the following parts.
[Part B: Time-averaged gravitational field of the Sun]

The Sun exerts a time-dependent torque on Earth, which can be replaced by its yearly averaged effect, modeled as a uniform mass ring of radius $d_{SE}$. ...

(B.1) Compute the gravitational field $g_z(z)$ on the $z$-axis due to the ring (for $|z| \ll d_{SE}$).

(B.2) Compute the gravitational field $g_r(r)$ in the ecliptic plane (for $r \ll d_{SE}$).

[Part C: Torque Acting on the Earth]

The Earth may be thought of as a perfect sphere of radius $R_e$ with two excess caps removed.

...

(C.1) Find the mass $m$ of one excess cap. (C.2) Using the effective two-point-mass model, compute the torque $\tau$ exerted by the solar ring.

[Part D: Angular Velocity of Earth's Axial Precession]

(D.1) Express the precession period $T_1$ in terms of $M_S$, $d_{SE}$, $\omega$, $h_{\max}$, $R$, and $\alpha$.

**Question:** Calculate the precession period $T_1$ in years.

**Final Answer:**

$$80600$$

**Source:** APhO 2025

---

**Model Answer (Qwen2.5-VL-7B-Instruct):**

...

**Part D: Angular Velocity of the Earth's Axial Precession**

**(D.1) Precession period $T_1$.**

The precession period is:

$$T_1 = \frac{4\pi^2 d_{SE}^3}{GM_S\omega \sin\alpha \cdot \frac{12}{25} \cdot \frac{h_{\max}}{R}}$$

Substituting the values:

$$T_1 \approx 25800 \text{ years}$$

$$\boxed{25800}$$

**Marking Scheme:** Award 0.2 pt if the answer gives the correct numerical result for the precession period as $T_1 \approx 80600$ years, obtained by correctly substituting the given data into a dimensionally correct formula. Partial points: award 0 pt if the substitution is incorrect or if the formula used has a dimensional error.

**Error Type:** Task Misunderstanding

**Comments:** The model was asked to calculate the precession period $T_1$ based on its derived formulas, but instead provided an empirical value (25800 years) directly from the problem's introduction, indicating a failure to perform the required calculation.

---

**Case 2 Concept & Computation Error - Lack of Physics Knowledge**

**Problem:**

**[Totality]**

Total solar eclipses are a rare phenomenon which occur when the Moon completely covers the disk of the Sun for some parts of the Earth. This doesn't happen during every solar eclipse because the Moon's apparent size in the sky is sometimes too small to fully cover the Sun, but also because the Moon's shadow usually misses the Earth due its orbital inclination. As a result, total solar eclipses occur on average every 18 months.

Let us consider a total solar eclipse where during the peak, the centre-points of Earth, the Moon and the Sun lie on a line on the same plane as the equator. We measure that right before the total solar eclipse ends at latitude $\lambda = 28.5°$, the totality lasts for $t_0 = 2\,\text{min}$. Earth's radius is $r_e = 6370\,\text{km}$, Moon's radius is $r_m = 1740\,\text{km}$, orbital period of the

Moon $T_m = 27.3\,\mathrm{d}$, orbital radius of the Moon $R_m = 384000\,\mathrm{km}$. One day on Earth is $T_0 = 24\,\mathrm{hrs}$.

(i) For how long is there a place on Earth where the total solar eclipse is observable?

(ii) How many degrees in longitude on Earth does the total solar eclipse cover?

Parts (i)–(ii) are preliminary questions and should not be included in the final answer.

**Question:** What is the width of the path of totality near the equator? Express your answer in $\mathrm{km}$.

**Final Answer:**

$$180$$

**Source:** NBPhO 2024

---

**Model Answer (LLaMA4-Scout-17B):**

The problem asks for the width of the path of totality near the equator during a total solar eclipse. Given data include $r_e = 6370\,\mathrm{km}$, $r_m = 1740\,\mathrm{km}$, $R_m = 384000\,\mathrm{km}$, $\lambda = 28.5°$, $t_0 = 2\,\mathrm{min}$, etc. ...

The model attempts to use geometric reasoning instead of the intended kinematic method. ...

The model writes several incorrect expressions for $w$:

$$w = 2\sqrt{r_m^2 - \left(\frac{r_m}{R_m} r_e \sin\lambda\right)^2} \frac{R_m}{r_e} r_e \cos\lambda \frac{v}{v}$$

$$w = 2 r_e \cos\lambda \sin\theta$$

$$w = 2 r_e \sqrt{\left(\frac{r_m}{R_m}\right)^2 - \left(\frac{r_e \sin\lambda}{R_m}\right)^2}$$

$$w = 2 r_e \frac{r_m}{R_m}$$

Substituting into the wrong formula:

$$w = 2 \times 6370 \times \frac{1740}{384000} \approx 57.75$$

The model outputs:

$$\boxed{57.75}$$

**Marking Scheme:** Award 0.4 pt if the answer realizes that the velocities are perpendicular at the end of the eclipse. Otherwise, award 0 pt.

**Error Type:** Lack of Physics Knowledge

**Comments:** The model uses an incorrect geometric formula to calculate the width of the path of totality, which does not represent the physics of the Moon's umbra. It also completely ignores the crucial information about the duration of totality ($t_0$), indicating a lack of understanding of the intended kinematic approach for this problem, as hinted by the marking scheme.

---

### Case 3 Concept & Computation Error - Calculation Mistakes

**Problem:**

**[Generation of ultrashort electromagnetic pulse]**

The Nobel Prize in Physics 2018 & 2023 were awarded to pioneers who contributed to "Method of generating high-intensity, ultra-short optical pulses" and "Generation of attosecond pulses of light for the study of electron dynamics in matter". Attosecond pulse refers to electromagnetic field with a duration on the order of $10^{-18}$ second. The advent of attosecond technique has made possible the study of ultrafast dynamics in physical, chemical and bio-

logical systems at a record high temporal resolution. Thus far, the most widely used method to generate attosecond pulse (Nobel Prize in Physics 2023) is to rely on the interaction of gas molecules and intensive femtosecond laser pulse (Nobel Prize in Physics 2018). In this question, we will explore some important aspects of the short pulse generation.
The following identity may be useful:

$$\int_{-\infty}^{\infty} e^{-a\omega^2} e^{-i\omega t} \, d\omega = \sqrt{\frac{\pi}{a}} \exp\left(-\frac{t^2}{4a}\right)$$

**Physical constants:** Electric charge: $e = 1.60 \times 10^{-19}\,\text{C}$
Electron mass: $m_e = 9.11 \times 10^{-31}\,\text{kg}$
Speed of light in vacuum: $c = 3.00 \times 10^8\,\text{m/s}$
Planck constant: $h = 6.63 \times 10^{-34}\,\text{J} \cdot \text{s}$

**Part D: High-harmonic generation of attosecond pulse**
Take a short femtosecond pulse from Ti:sapphire laser (center frequency $\omega_0$) and focus it into a gas medium. One could generate light at integer multiples of the driving frequency, often referred to as high-harmonic generation (HHG).
*One critical step for high-harmonic generation (HHG) is to drive the bound charge of molecules in the non-perturbative regime to initiate the ionization dynamics. If we take hydrogen molecules for HHG, please estimate the peak electric field strength required to initiate the process (expressed in $E >?\,\text{GV/m}$).*
**Final Answer:**
$$E > 30 \, fracGVm$$

**Source:** PanPhO 2025

---

**Model Answer (GPT-5):**
To reach the non-perturbative (tunneling) regime that initiates ionization in HHG, set the Keldysh parameter $\gamma \approx 1$:

$$\gamma = \frac{\omega_0 \sqrt{2m_e I_p}}{eE} \quad \Rightarrow \quad E \approx \frac{\omega_0 \sqrt{2m_e I_p}}{e}.$$

Using Ti:sapphire laser ($\lambda = 800\,\text{nm} \Rightarrow \omega_0 = \frac{2\pi c}{\lambda} \approx 2.36 \times 10^{15}\,\text{s}^{-1}$) and hydrogen molecule ionization potential $I_p \approx 15.4\,\text{eV} = 2.47 \times 10^{-18}\,\text{J}$, we get:

$$E \approx \frac{(2.36 \times 10^{15}) \cdot \sqrt{2 \cdot 9.11 \times 10^{-31} \cdot 2.47 \times 10^{-18}}}{1.60 \times 10^{-19}} \approx 3.1 \times 10^{10}\,\text{V/m} = 31\,\text{GV/m}.$$

Thus, the onset field is on the order of a few $\times\,10\,\text{GV/m}$.

$$\boxed{30}$$

**Error Type:** Calculation Mistakes

**Comments:** The model correctly calculates the electric field strength to be approximately 31 GV/m based on the Keldysh parameter. However, its final boxed answer is 30, indicating a minor rounding discrepancy or an arbitrary adjustment to the numerical output.

---

### *Case 4 Physics Reasoning Error - Misapplication of Physical Laws*

**Problem:**
**[Cox's Timepiece]**
In 1765, James Cox invented a clock powered by atmospheric pressure fluctuations, using mercury transfer between two vessels. This analysis focuses on a two-part barometric tube immersed in a mercury bath:
- Tube (lower): area $S_t$, height $H_t = 80$ cm;

- Bulb (upper): area $S_b > S_t$, height $H_b = 20$ cm.

The system contains no air and is subject to gravity $g = 9.8$ m/s$^2$, with mercury density $\rho = 13.5 \times 10^3$ kg/m$^3$. The force to maintain equilibrium is:

$$\vec{F} = (m_{\text{tb}} + m_{\text{add}})\, g\vec{u}_z$$

**(B.1)** Describe the region responsible for $m_{\text{add}}$. **(B.2)** With fixed pressure $P_0 = 10^5$ Pa, sketch the evolution of $m_{\text{add}}$ as a function of $h_t \in [-H_b, H_t]$:

- Give slopes of each segment;
- Identify coordinates of angular points using $P_0, \rho, g, S_b, S_t, H_b, H_t$.

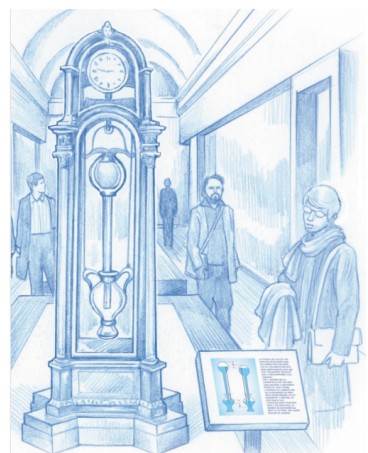

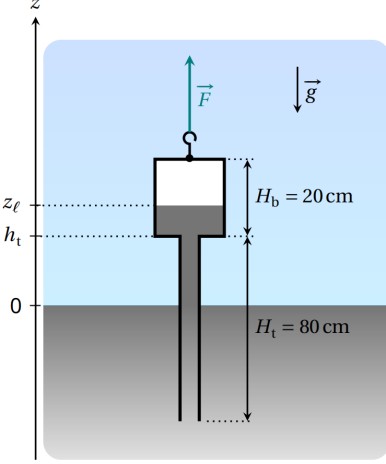

**Fig. 1.** Artistic view of Cox's clock [1]       **Fig. 3.** Sketch of the two-part barometric tube

**Final Answer:**

$$\text{Segment 1: slope } = \rho S_{\text{b}}$$
$$\text{Segment 2: slope } = \rho S_{\text{t}}$$
$$\text{Segment 3: slope } = -\rho(S_{\text{b}} - S_{\text{t}})$$
$$\text{Segment 4: slope } = 0$$

$$\text{Angular point 1: } 0$$
$$\text{Angular point 2: } z_\ell^\star - H_{\text{b}}$$
$$\text{Angular point 3: } z_\ell^\star$$

**Source:** IPhO 2025

---

**Model Answer (Gemini-2.5-Pro):**

...
**Regime 1: $h_t \leq h_0 - H_b$** (Tube is full) ... The slope is $\frac{dm_{\text{add}}}{dh_t} = \rho(S_t + S_b)$. ...
**Regime 2: $h_0 - H_b < h_t \leq h_0$** (Vacuum forms, mercury in bulb) ... The slope is $\frac{dm_{\text{add}}}{dh_t} = -\rho(S_b - S_t)$. ...
**Regime 3: $h_t > h_0$** ... The slope is $\frac{dm_{\text{add}}}{dh_t} = 0$. ...

**Marking Scheme:** Award 0.1 pt if the answer correctly gives the first two slopes: slope = $\rho S_{\text{b}}$ for Segment 1, and slope = $\rho S_{\text{t}}$ for Segment 2. If any one slope is incorrect, award 0 pt.

**Error Type:** Misapplication of Physical Laws

**Comments:** The model's derived functional form of $m_{\text{add}}(h_{\text{t}})$ is incorrect, leading to erroneous slopes for all active segments. This stems from algebraic errors in combining terms and/or misidentifying how relevant physical quantities change with $h_{\text{t}}$.

*Case 5 Physics Reasoning Error - Faulty Symbolic Derivation*

**Problem:**
考虑两个质量分别为 $m$ 和 $M$ 的物体在光滑水平面上作一维运动并发生弹性碰撞。$m$ 和 $M$ 的初速度分别为 $u$ 和 $U$，碰撞后的速度分别为 $v$ 和 $V$。
已知 $m < M$ 且 $m$ 最初处于静止状态，而 $M$ 在右侧以速度 $U$ 向 $m$ 移动。将向右的方向视为正方向，因此 $U < 0$。初次碰撞后，$m$ 向左移动并在一定距离处撞到一墙壁，然后反弹回来再次向右移动。然后它可能再次撞到 $M$ 并再反弹回来撞到墙壁。将 $m$ 和 $M$ 之间的初次碰撞称为第 0 次碰撞。我们假设所有碰撞都是弹性碰撞。设 $m$ 和 $M$ 在第 $n$ 次碰撞之前的速度分别为 $v_n$ 和 $V_n$，因此 $V_0 = U$ 和 $v_0 = 0$。
用相空间中的一个点 $(\sqrt{M}V, \sqrt{m}v)$ 表示两个物体的运动状态。因此，在第一次碰撞之前，状态为 $(\sqrt{M}U, 0)$。在第二次碰撞之前，状态为 $(\sqrt{M}V_1, \sqrt{m}v_1)$。
(1) 求 $v_n$ 的表达式，用 $m, M,$ 和 $U$ 表示。
(2) 求 $V_n$ 的表达式，用 $m, M,$ 和 $U$ 表示。

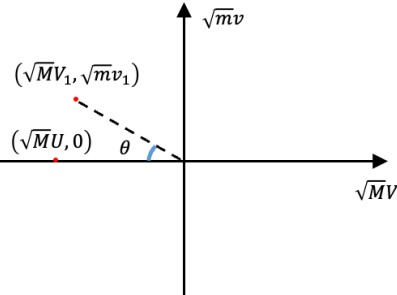

**Final Answer:**
$$v_n = -\sqrt{\frac{M}{m}}\, U \sin\left(n \tan^{-1} \frac{2\sqrt{Mm}}{M-m}\right), \quad V_n = U \cos\left(n \tan^{-1} \frac{2\sqrt{Mm}}{M-m}\right)$$

**Source:** PanMechanics 2025

**Model Answer (GLM-4.5V):**
... 通过分析发现，每次碰撞（包括 $m$ 与墙壁的碰撞）对应相空间中的一个线性变换。具体地，碰撞后相空间坐标可通过以下矩阵变换表示：
$$\begin{pmatrix} \sqrt{M}V_{n+1} \\ \sqrt{m}v_{n+1} \end{pmatrix} = \begin{pmatrix} \frac{M-m}{M+m} & \frac{2m}{M+m} \\ \frac{2M}{M+m} & \frac{m-M}{M+m} \end{pmatrix} \begin{pmatrix} \sqrt{M}V_n \\ \sqrt{m}v_n \end{pmatrix}.$$
...

**Error Type:** Faulty Symbolic Derivation

**Comments:** The model incorrectly formulates the phase space transformation matrix for an m-M collision (e.g., using $2m$ instead of $2\sqrt{Mm}$). Crucially, it completely omits the transformation due to the m-wall collision, which is a necessary step between successive m-M collisions, leading to an incorrect recurrence relation for the system's evolution.

*Case 6 Physics Reasoning Error - Dimensional Analysis Lapses*

**Problem:**
You are asked to study the features of the brightly lit circle and dark rings in the figures below. Make your calculations for an idealized situation: the chair leg is strictly cylindrical of radius $a$, strictly vertical, with a perfectly smooth, cylindrical, and perfectly reflecting surface. You may make any additional model assumptions and approximations you deem reasonable that will simplify your calculations.
Determine how the illuminance surplus $I(r, \theta)$ inside the brightly lit circle on the floor depends on the polar coordinates $r \gg a$ and $\theta$. The illuminance quantifies the amount of incoming light per area. By "surplus" we mean the additional illuminance introduced due to

the presence of the cylinder. Express the answer in terms of $I_0$, defined as the illuminance difference between points $A$ and $B$ in the figure.

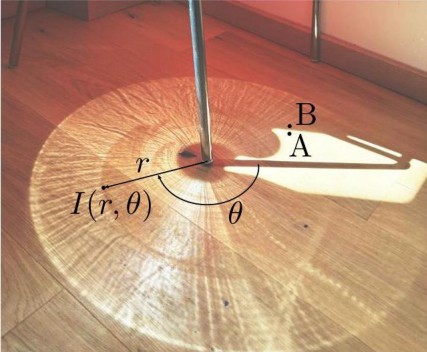

**Final Answer:**

$$I(r, \theta) = \frac{I_0 a}{2r} \sin\left(\frac{\theta}{2}\right)$$

**Source:** EuPhO 2025

---

**Model Answer (InternVL3-78B-Instruct):**

...

Given the lack of progress, I'll settle on the answer $I(r, \theta) = \frac{I_0 \cos \theta}{r}$ , acknowledging that this is an educated guess based on typical angular dependencies in optics problems.

...

> **Marking Scheme:** Award 0.5 pts if the answer gives $I = -\frac{I_0 a}{2l} \cos(\beta)$.

**Error Type:** Dimensional Analysis Lapses

**Comments:** The model's final expression $\frac{I_0 \cos \theta}{r}$ has units of illuminance per unit length (e.g., Lux/meter) because $I_0$ is an illuminance and $r$ is a length. However, the illuminance surplus $I(r, \theta)$ should have units of illuminance (Lux), indicating a dimensional inconsistency where a length factor (like $a$ from the correct answer) is missing in the numerator.

---

### *Case 7 Visual Reasoning Error - Scene Misinterpretation*

**Problem:**

**[Table and Small Oscillations]**

A table is made by fastening a metal frame to a massive uniform plate (so they form a rigid body) and attaching it with chains to another frame that is fixed on the horizontal ground. The motion of the table is limited to the plane of the side view (right picture).

The masses of the chains and the frame can be neglected. The chains are frictionless, inextensible, and remain tensioned in oscillations. The grid step is $a = 0.100$ m, and the acceleration due to gravity is $g = 9.81$ m/s$^2$.

**Question:** Find the period $T$ of the small oscillations:
(1) Write the formula for $T$
(2) Calculate the value of $T$ (keep three significant figures and express the unit in seconds).

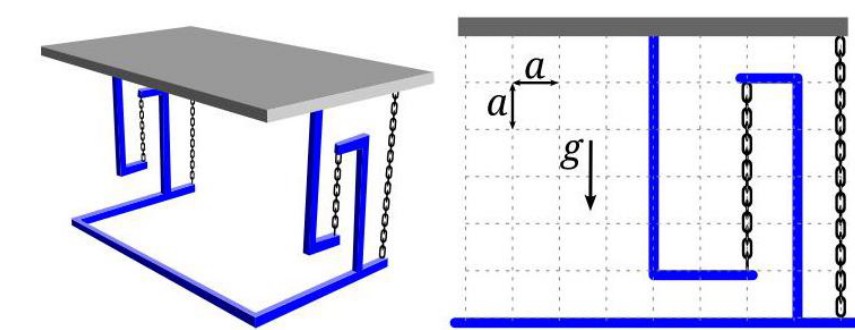

**Final Answer:**

$$T = 2\pi\sqrt{\frac{3a}{2g}}, \quad T = 0.777 \text{ s}$$

**Source:** EuPhO 2025

---

**Model Answer (Gemini-2.5-Flash-Thinking):**

...

Let's assume the most common interpretation of such diagrams: The blue frame is a rigid body. The tabletop is also a rigid body. They are fastened together to form a larger rigid body. The *mass* of this rigid body is only the mass of the tabletop, $M$. The CM of this rigid body is the CM of the tabletop: $C = (4a, 7.5a)$. The moment of inertia of this rigid body about its CM is $I_{CM} = \frac{65}{12}Ma^2$.

Now, how is this rigid body attached to the fixed ground? "attaching it with chains to another frame that is fixed on the horizontal ground." The "fixed frame on the horizontal ground" is the bottom blue line. The chains are drawn explicitly on the left and right, connecting the tabletop to the ground. Left chain: from (0, 7a) to (0, 0). Right chain: from (8a, 7a) to (8a, 0).

These chains are vertical. If the table moves horizontally, it would move parallel to itself. Let the table move by a small angle $\theta$ around some pivot. If the chains are vertical, they don't provide a restoring torque for angular displacement. If the table translates horizontally, there is no restoring torque.

This means the system cannot oscillate angularly, contradicting the setup of a physical pendulum.

...

> **Marking Scheme:** Award 2.0 pts if the answer includes all necessary elements: small horizontal forces; correct approximations of kinetic and potential energies; accelerations/curvatures. Partial points: Award 1.0 pt if not all elements are present or if they contain mistakes; award 1.0 pt if the proof that $\omega_0 = 0$ is missing (or rotation is not considered), but everything else is correct; award 0.0 pts if only partial elements are present with an unrelated approach (e.g., using energy but only writing force expressions).

> **Error Type:** Scene Misinterpretation

> **Comments:** The model failed to correctly interpret the physical scene and geometry from the ambiguous diagram and problem description. This inability to define the oscillating rigid body and its connection to the fixed ground (e.g., pivot points) prevented it from formulating a physical model and attempting a solution.

### Case 8 Visual Reasoning Error - Data-reading Inaccuracy

**Problem:**

**[Hydrogen and galaxies]**

This problem aims to study the peculiar physics of galaxies, such as their dynamics and structure. In particular, we explain how to measure the mass distribution of our galaxy from

the inside. For this we will focus on hydrogen, its main constituent.

Throughout this problem we will only use $\hbar$, defined as $\hbar = h/(2\pi)$.

**[Part B - Rotation curves of galaxies]**

**Data:**

Kiloparsec: 1 kpc $= 3.09 \times 10^{19}$ m

Solar mass: 1 $M_\odot = 1.99 \times 10^{30}$ kg

We consider a spherical galaxy centered around a fixed point $O$. At any point $P$, let $\rho = \rho(P)$ be the volumetric mass density and $\varphi = \varphi(P)$ the associated gravitational potential (i.e. potential energy per unit mass). Both $\rho$ and $\varphi$ depend only on $r = \|\vec{OP}\|$. The motion of a mass $m$ located at $P$, due to the field $\varphi$, is restricted to a plane containing $O$.

(B.1) In the case of a circular orbit, determine the velocity $v_c$ of an object on a circular orbit passing through $P$ in terms of $r$ and $\frac{d\varphi}{dr}$.

Fig. 1(A) is a picture of the spiral galaxy NGC 6946 in the visible band (...). The little ellipses in Fig. 1(B) show experimental measurements of $v_c$ for this galaxy. The central region ($r < 1$ kpc) is named the bulge. In this region, the mass distribution is roughly homogeneous. The red curve is a prediction for $v_c$ if the system were homogeneous in the bulge and keplerian ($\varphi(r) = -\beta/r$ with $\beta > 0$) outside it, i.e. considering that the total mass of the galaxy is concentrated in the bulge.

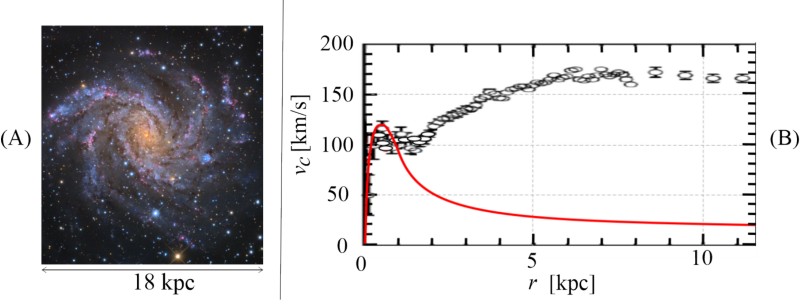

**Fig. 1**: NGC 6946 galaxy: Picture (A) and rotation curve (B).

**Final Answer:**

$$[6.75 \times 10^8, 11.25 \times 10^8]$$

**Source:** IPhO 2025

---

**Model Answer (Claude-4-Sonnet-Thinking):**

At the boundary of the bulge ($r = 1$ kpc), I need to read the velocity from the red curve in Fig. 1(B). Looking at the red curve at $r = 1$ kpc, the velocity appears to be approximately 125 km/s. ... $M_b = \frac{(1.25 \times 10^5)^2 \times 3.09 \times 10^{19}}{6.674 \times 10^{-11}}$ ... Converting to solar masses: $M_b = \frac{7.23 \times 10^{39}}{1.99 \times 10^{30}} = 3.6 \times 10^9 M_\odot$

> **Marking Scheme:**
> Award 0.1 pt if the answer gives the expression for $\vec{g}(r > r_b)$ as $\vec{g}(r > r_b) = -\frac{GM_b}{r^2}\vec{u}_r$ with $r > r_b$ and $M_{\text{int}}(r) = M_b$. Otherwise, award 0 pt.

> **Error Type:** Data-reading Inaccuracy

> **Comments:** The model correctly identified the formula for Keplerian motion outside the bulge and applied it at the boundary ($r = 1$ kpc). However, the velocity read from the red curve at this point (approximately 125 km/s) leads to a bulge mass significantly higher than the accepted answer. This suggests that while the visual reading was accurate for the diagram, this specific data point was not the intended one to deduce the bulge mass or the diagram is inconsistent with the expected answer. Therefore, the error is in selecting the wrong data point for calculation, leading to an incorrect result.

### Case 9 Visual Reasoning Error - Variable Misunderstanding

**Problem:**

**[Harrison and the Longitude Problem]**

Accurate measurement of the longitude was a long-standing problem in sea navigation. The earliest solution to this problem was to compare the local time at a location with that of a meridian. However, there was not a clock accurate enough to preserve the absolute time of another location during a journey, until the inventions by the 18$^{th}$ century English clockmaker John Harrison.

Let us start our discussion on the simplest type of clock, which is maintained by a vertically hung simple pendulum. It is made of a heavy bob of mass $m$, hung by a rod of length $L$ and negligible mass, to a hinge. The timekeeping is evidently sensitive to temperature fluctuations.

**Question:** In adjustment, Harrison proposed a modification to the rod of a pendulum, now consisting of two types of metals (of thermal expansion coefficients $\alpha_1$ and $\alpha_2$ respectively) by installing a central piece of metal 2 of length $l'$.

(1) Which of the 3 proposals in Fig. 4(a) can suppress the temperature fluctuations?

(2) How long should the middle piece $l'$ of metal 2 be such that the total length of the rod $L$ is independent of temperature fluctuations in this proposal? Express your answer in terms of $L$, $\alpha_1$ and $\alpha_2$.

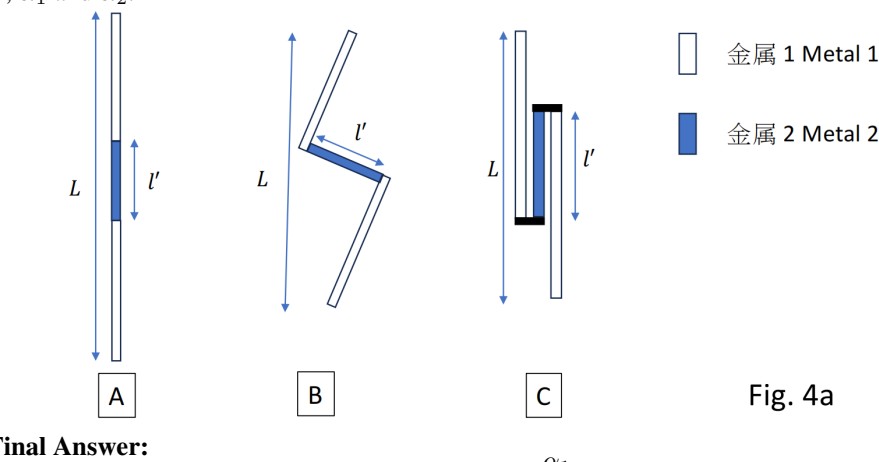

金属 1 Metal 1

金属 2 Metal 2

Fig. 4a

**Final Answer:**

$$(1)\ B, \quad (2)\ l' = \frac{\alpha_1}{\alpha_2 - \alpha_1}L$$

**Source:** PanPhO 2025

---

**Model Answer (DeepSeek-R1):**

...

To proceed, in the boxed answer, for (1) I'll put the choice, but since not specified, perhaps the problem expects only the expression, but it asks for both.

... so for the answer, I can put $\boxed{C}$ for (1), and for (2), the expression.

...

Final boxed answer: $\left[\ \boxed{C}\ ,\ \boxed{\dfrac{L\alpha_1}{\alpha_2 - \alpha_1}}\ \right]$

**Error Type:** Variable Misunderstanding

**Comments:** The model correctly derives the formula for thermal compensation but incorrectly selects proposal 'C' instead of the correct 'B' for the compensating mechanism, likely due to the absence of 'Fig. 4a' in the prompt. This indicates a misidentification of the correct proposal's label.

