# OpenReview forum: "HiPhO: How Far Are (M)LLMs from Humans in the Latest High School Physics Olympiad Benchmark?"
_ICLR.cc/2026/Conference — Submitted to ICLR 2026_

### Official Review · Reviewer_FQst · 2025-10-28

**Soundness:** 3
**Presentation:** 4
**Contribution:** 3
**Rating:** 4
**Confidence:** 5

**Summary:**

This paper introduces HiPhO, a benchmark for evaluating MLLMs on recent high-school physics Olympiad exams. It emphasizes: (1) up-to-date coverage of 13 Olympiad exams; (2) human-aligned evaluation that combines answer-level checks with step-level scoring using official marking schemes; and (3) direct comparison with human contestants via medal thresholds. Large-scale experiments on 30 models show closed-source reasoning MLLMs frequently reach gold-level scores but still trail top human contestants, while open-source MLLMs generally remain bronze-level; strong open-source LLMs (text-only) can earn multiple golds on text-dominant tasks.

**Strengths:**

1. This benchmark covers 13 Olympiads, 5 different fields, and 4 modality types. This is the most comprehensive benchmark for physics olympic competition.

2. Human-aligned evaluation with official marking schemes and step-level score is more faithful than answer-only accuracy.

3. This paper demonstrates insightful conclusions, including modality difficulty, field-wise gaps, and visual challenges.

4. The writing is good and easy to follow.

**Weaknesses:**

1. This benchmark does not evaluate the training data contamination of the models. It is highly likely that closed-source MLLMs have already been trained on these competition problems, since they are publicly available. Providing contamination results would better support the credibility and validity of this benchmark.

2. Although the authors emphasize the modality/field gaps, they could further include fine-grained failure modes (e.g., dimensional analysis lapses, algebraic slips, diagram parsing errors, data-reading inaccuracies) with concrete examples and per-type confusion patterns to better guide method development. If a subset of data were sampled for manual error analysis, it could more deeply reveal the challenges specific to this domain.

**Questions:**

No

---

> ### Author Response · Authors · 2025-11-27
>
> Thank you for recognizing our work as the most comprehensive benchmark for physics Olympiads, featuring human-aligned evaluation, insightful conclusions, and clear writing. **As recommended, we have revised the paper and added data leakage checks and fine-grained error analysis.**
>
> ---
>
> **Comment 1.** This benchmark does not evaluate the training data contamination of the models. It is highly likely that closed-source MLLMs have already been trained on these competition problems, since they are publicly available. Providing contamination results would better support the credibility and validity of this benchmark.
>
> **Response 1.** To reduce data-leakage risks, we followed OlympiadBench [1] and sourced all problems from official websites and converted them from PDF to markdown, reducing the chance that these items appear in model pre-training data. We also used only the latest 2024-2025 Olympiads to avoid temporal overlap; for example, APhO 2025 was released on August 16, 2025, after all evaluated models.
>
> We further conducted explicit data‑leakage checks. Following the **partial‑prompt completion** strategy in [2], we randomly sampled 90 problems (25% benchmark data), ensuring coverage across all Olympiads. For each problem, we provided the first 60% of the content and asked the model to greedily complete the remainder. We then computed the same metrics as in [2]:
>
> - **ROUGE-L (Partial-Prompt Completion Rate):** Measures the overlap of the longest common subsequence between the generated and reference text.
> - **ACC (Partial-Prompt Answer Accuracy):** Measures the proportion of problems where the generated text contains the correct answer.
>
> As shown in **Table R1**, we evaluated 10 top-performing models and found that their completion rates are all below **15%**, with answer accuracy below **4%** on average. The detailed results in **Tables R2 & R3** further confirm that across all 13 Olympiads, most completion rates remain below **20%** and most answer accuracies remain at **0%**. These findings indicate that the HiPhO benchmark exhibits **minimal data leakage** and remains **highly challenging**. We have included these data leakage checks in **Section 3.4** and **Appendix B.3**.
>
> **Table R1. Summary of Data Leakage Check (%) on the HiPhO subset.**
> |Closed-source Models|ROUGE-L|ACC|Open-source Models|ROUGE-L|ACC|
> -|-|-|-|-|-
> Gemini-2.5-Pro|11.8|1.7|Intern-S1|10.9|2.2
> Gemini-2.5-Flash-Thinking|11.8|1.7|InternVL3-78B-Instruct|12.2|1.1
> GPT-5|12.1|3.7|Qwen2.5-VL-72B-Instruct|10.8|1.1
> Grok-4|14.9|1.7|DeepSeek-R1|10.3|0.0
> Claude-4-Sonnet-Thinking|10.7|1.1|Qwen3-235B-A22B|8.4|1.1
>
> ---
>
> **Table R2. Partial-prompt completion rate (\%) across all Olympiads.**
> |ROUGE-L|IPhO||APhO|EuPhO||NBPhO||PanPhO||PanMechanics||F=MA||
> |-|-|-|-|-|-|-|-|-|-|-|-|-|-|
> ||2025|2024|2025|2025|2024|2025|2024|2025|2024|2025|2024|2025|2024|
> |*Closed-source Models*||||||||||||||
> |Gemini-2.5-Pro|15.9|15.2|14.9|6.5|14.3|17.2|15.9|13.3|13.9|4.1|5.1|6.1|4.9|
> |Gemini-2.5-Flash-Thinking|11.5|11.4|10.0|13.5|18.5|7.2|7.6|12.6|11.2|6.9|7.4|16.2|19.3|
> |GPT-5|11.2|8.0|13.3|11.4|11.1|15.1|12.0|13.8|11.9|5.6|8.6|15.7|20.2|
> |Grok-4|12.2|10.9|13.4|15.0|22.1|15.7|14.9|16.0|13.1|5.1|10.3|18.2|29.3|
> |Claude-4-Sonnet-Thinking|11.5|14.7|12.9|10.5|7.1|12.3|11.5|13.5|13.6|3.7|5.9|8.4|7.3|
> |*Open-source Models*||||||||||||||
> |Intern-S1|12.9|13.6|13.4|9.1|12.7|11.0|13.7|13.4|12.5|2.5|3.5|7.9|10.0|
> |InternVL3-78B-Instruct|11.9|14.0|19.1|6.9|12.3|12.5|10.6|13.2|14.0|14.7|10.6|7.6|6.4|
> |Qwen2.5-VL-72B-Instruct|12.5|12.4|18.2|10.3|11.1|11.2|10.3|12.4|12.2|3.0|7.4|4.6|7.5|
> |DeepSeek-R1|13.0|13.3|12.0|6.6|12.5|13.1|10.5|10.2|13.1|1.3|5.7|8.2|9.8|
> |Qwen3-235B-A22B|12.0|13.8|10.9|7.9|9.7|8.0|8.3|7.7|11.5|0.0|1.5|6.5|5.8|
>
> ---
>
> **Table R3. Partial-prompt answer accuracy (\%) across all Olympiads.**
> |ACC|IPhO||APhO|EuPhO||NBPhO||PanPhO||PanMechanics||F=MA||
> |-|-|-|-|-|-|-|-|-|-|-|-|-|-|
> ||2025|2024|2025|2025|2024|2025|2024|2025|2024|2025|2024|2025|2024|
> |*Closed-source Models*|||||||||||||||
> |Gemini-2.5-Pro|0.0|0.0|0.0|0.0|8.3|0.0|0.0|9.1|0.0|0.0|0.0|0.0|0.0|
> |Gemini-2.5-Flash-Thinking|0.0|0.0|0.0|0.0|8.3|0.0|0.0|9.1|0.0|0.0|0.0|0.0|0.0|
> |GPT-5|11.1|0.0|12.5|0.0|0.0|0.0|0.0|12.1|0.0|0.0|0.0|0.0|0.0|
> |Grok-4|11.1|0.0|0.0|0.0|8.3|0.0|0.0|0.0|0.0|0.0|0.0|0.0|0.0|
> |Claude-4-Sonnet-Thinking|0.0|0.0|0.0|0.0|0.0|0.0|0.0|0.0|0.0|14.3|0.0|0.0|0.0|
> *Open-source Models*||||||||||||||
> |Intern-S1|11.1|0.0|0.0|0.0|0.0|0.0|0.0|0.0|0.0|14.3|0.0|0.0|0.0|
> |InternVL3-78B-Instruct|0.0|0.0|0.0|0.0|0.0|0.0|0.0|0.0|0.0|0.0|0.0|0.0|16.7|
> |Qwen2.5-VL-72B-Instruct|0.0|0.0|0.0|0.0|0.0|0.0|0.0|0.0|0.0|14.3|0.0|0.0|0.0|
> |DeepSeek-R1|0.0|0.0|0.0|0.0|0.0|0.0|0.0|0.0|0.0|0.0|0.0|0.0|0.0|
> |Qwen3-235B-A22B|11.1|0.0|0.0|0.0|0.0|0.0|0.0|0.0|0.0|0.0|0.0|0.0|0.0|
>
> [1] Olympiadbench: A challenging benchmark for promoting agi with olympiad-level bilingual multimodal scientific problems. ACL 2024.
>
> [2] Reasoning or memorization? unreliable results of reinforcement learning due to data contamination. arXiv 2507.

---

> ### Author Response · Authors · 2025-11-27
>
> **Comment 2.** Although the authors emphasize the modality/field gaps, they could further include fine-grained failure modes (e.g., dimensional analysis lapses, algebraic slips, diagram parsing errors, data-reading inaccuracies) with concrete examples and per-type confusion patterns to better guide method development. If a subset of data were sampled for manual error analysis, it could more deeply reveal the challenges specific to this domain.
>
> **Response 2.** As recommended, we further performed a fine-grained error analysis on 85 problems from the HiPhO benchmark where both Gemini-2.5-Pro and GPT-5 scored $\leq$75% of the total points. For each step-level criterion marked incorrect, we assign one of nine error subtypes under three main categories:
> - **Concept & Computation Error:** These errors stem from missing conceptual understanding or simple numerical mistakes, including *Task Misunderstanding*, *Lack of Physics Knowledge*, and *Calculation Mistakes*.
> - **Physics Reasoning Error:** These involve incorrect symbolic or physical reasoning, including *Misapplication of Physical Laws*, *Faulty Symbolic Derivation*, and *Dimensional Analysis Lapses*.
> - **Visual Reasoning Error:** These arise from misinterpreting diagrams or data, including *Scene Misinterpretation*, *Data-reading Inaccuracy*, and *Variable Misunderstanding*.
>
> As shown in **Fig. 7** (**Section 5.3**), our findings show that:
>
> - **Open-source models make notably more errors, especially in physics knowledge and law application**, indicating substantial room for improvement in domain understanding.
> - **Closed-source models mainly struggle with task misinterpretation and symbolic derivation**, underscoring the complexity of Olympiad-style multi-step reasoning.
> - **Scene misinterpretation is the most frequent visual error for open-source models**, highlighting persistent challenges with interpreting physical setup in figures.
>
> These patterns expose key weaknesses in both physics and visual reasoning. **Representative examples of each error type are provided in Appendix F.**

---

### Official Review · Reviewer_tc7S · 2025-10-31

**Soundness:** 3
**Presentation:** 3
**Contribution:** 3
**Rating:** 6
**Confidence:** 4

**Summary:**

This paper introduces HiPhO, a new benchmark designed to evaluate the physical reasoning capabilities of large language models (LLMs) and multimodal large language models (MLLMs) using high school physics Olympiad problems. The benchmark's main contributions are threefold: (1) a comprehensive and up-to-date dataset of 13 international and regional Olympiad exams from 2024–2025, featuring mixed-modality problems; (2) a professional, human-aligned evaluation methodology that uses official marking schemes for both final-answer and step-level grading; and (3) a direct comparison with human contestants by mapping model scores to official gold, silver, and bronze medal thresholds. The authors evaluate 30 state-of-the-art models, finding that top closed-source models achieve gold-level performance in many exams but still fall short of the absolute top human scores. The results also highlight a significant performance gap between leading models and the broader field of open-source models, particularly open-source MLLMs.

**Strengths:**

*   **Originality & Significance:** The most significant contribution of this work is its holistic, human-aligned evaluation framework. While other physics benchmarks exist, HiPhO is the first to systematically benchmark models against real-world human performance standards (medal cutoffs) from recent competitions.

*   **Quality:** The quality of the benchmark and the evaluation is high.

*   **Clarity:** The paper is written with clarity. The motivation is well-established, the contributions are clearly articulated, and the structure is logical and easy to follow.

**Weaknesses:**

*   **Reliance on a Proprietary Grader Model:** The step-level evaluation, a key contribution, relies entirely on API calls to Gemini-2.5-Flash. While the authors compellingly argue for its superiority over GPT-4o, this introduces a dependency on a closed-source model that is subject to change, deprecation, or cost fluctuations. This presents a long-term risk to the benchmark's reproducibility and accessibility. While a difficult problem to solve, the authors could strengthen the paper by discussing this limitation and suggesting future work toward creating a fully open-source, fine-tuned grader to accompany the benchmark.

*   **Anecdotal Evidence for Grader Choice:** Table 7, which provides crucial evidence for the choice of grader, is based on a "single inference run example." For a claim as central as the quality of the evaluation metric, this is anecdotal. A more robust analysis would compare the grader models' outputs to human expert scores across a larger sample of problems and multiple runs to provide statistical evidence (e.g., correlation, mean absolute error) of the grader's reliability.

**Questions:**

1.  **On the Impact of Step-Level Grading:** Your evaluation metric is `max(answer-level score, step-level score)`. This is a sensible choice. Could you provide a brief quantitative analysis of how frequently the step-level score was the decisive factor (i.e., when a model got the final answer wrong but still received partial credit for correct steps)? This would provide powerful evidence for the necessity of your more fine-grained evaluation method.

2.  **On Medal Threshold Determination:** For Olympiads with both theory and experimental parts (IPhO, EuPhO), you set the medal threshold based on the "lowest theoretical exam score" of human medalists. This seems reasonable, but could it be skewed by a contestant who scored extremely high on the experimental exam and just high enough on the theory exam to secure a medal? How sensitive are the medal thresholds to such outliers, and does this methodology create a robust and representative performance bar?

3.  **On Future Directions for Experimental Physics:** You correctly identify the inability to perform experiments as a key limitation. Beyond simply stating this, could you elaborate on what you believe are the most critical research steps needed to bridge this "embodiment" gap? For example, would this involve creating simulated physics environments, developing agents capable of active experimentation, or something else entirely? A more detailed perspective would add valuable depth to the future work discussion.

---

> ### Author Response · Authors · 2025-11-27
>
> Thank you for recognizing our contributions to a human-aligned evaluation framework, a high-quality benchmark, and clear writing. As suggested, we have revised the paper and added further analysis of the grader models.
>
> ---
>
> **Comment 1. Reliance on a Proprietary Grader Model:** While the authors compellingly argue for its superiority over GPT-4o, this introduces a dependency on a closed-source model that is subject to change, deprecation, or cost fluctuations. This presents a long-term risk to the benchmark's reproducibility and accessibility. While a difficult problem to solve, the authors could strengthen the paper by discussing this limitation and suggesting future work toward creating a fully open-source, fine-tuned grader to accompany the benchmark.
>
> **Response 1.** Thank you for pointing this out. Many recent physics benchmarks rely on proprietary graders, e.g., GPT‑4o [1,2,3]. However, our empirical analysis shows that the widely used GPT‑4o grader introduces noticeable discrepancies compared with human experts, motivating our choice of Gemini-2.5-Flash as a more reliable alternative for HiPhO.
>
> As suggested, we conducted an additional experiment across four graders: **Gemini‑2.5‑Flash** (used in our work), **DeepSeek‑R1** (open-source), **GPT‑4o** (commonly used), and **human experts**, across both IPhO 2025 and IPhO 2024. For each of three evaluated models, we independently graded three inference runs using all four graders. As shown in **Table R1** (**Appendix E.3**), Gemini‑2.5‑Flash consistently produces scores closely aligned with human experts (typically within 1 point). **DeepSeek‑R1 also performs well and may serve as an open-source alternative.** In contrast, GPT‑4o shows significant deviation, often underestimating model performance, especially on IPhO 2024.
>
> **Table R1. Comparison of Different Graders with Human Experts.**
> |Olympiad|IPhO 2025||||IPhO 2024||||
> -|-|-|-|-|-|-|-|-
> |Grader|Gemini-2.5-Flash|DeepSeek-R1|GPT-4o|Human|Gemini-2.5-Flash|DeepSeek-R1|GPT-4o|Human|
> Gemini-2.5-Pro|22.1|21.9|21.5|22.0|25.7|24.0|21.1|26.0
> Gemini-2.5-Flash-Thinking|19.9|20.0|20.4|19.1|23.5|22.5|20.0|23.3
> Claude-4-Sonnet-Thinking|19.3|20.2|18.5|19.0|22.1|21.0|17.4|21.3
>
> This provides an important insight from our study: **the choice of grader plays a critical role in benchmark evaluation but is often overlooked**. Instead of relying on a commonly used grader, we carefully examined the reliability of different graders. Although grading is not the primary focus of this paper, we have included relevant results and discussion in **Section 5.4** and **Appendix E.3**. In **Section 6**, our future work will develop an open-source grader to strengthen the benchmark.
>
> [1] UGPhysics: A Comprehensive Benchmark for Undergraduate Physics Reasoning with Large Language Models. ICML 2025.
>
> [2] PHYSICS: Benchmarking Foundation Models on University-Level Physics Problem Solving. arXiv 2503.
>
> [3] PhysicsArena: The First Multimodal Physics Reasoning Benchmark Exploring Variable, Process, and Solution Dimensions. arXiv 2505.
>
> ---
>
> **Comment 2. Anecdotal Evidence for Grader Choice:** Table 7, which provides crucial evidence for the choice of grader, is based on a "single inference run example." For a claim as central as the quality of the evaluation metric, this is anecdotal. A more robust analysis would compare the grader models' outputs to human expert scores across a larger sample of problems and multiple runs to provide statistical evidence (e.g., correlation, mean absolute error) of the grader's reliability.
>
> **Response 2.** As suggested, we have conducted a deeper analysis of different graders across both IPhO 2025 and 2024. We evaluated three models, each with three independent inference runs, and graded all outputs using four graders: Gemini-2.5-Flash, DeepSeek-R1, GPT-4o, and human experts. As shown in **Table R1** (Response 1), Gemini-2.5-Flash produce scores that are consistently closer to those of human experts, while GPT-4o shows larger deviations.
>
> To further quantify grader reliability, we conducted a statistical analysis of score consistency using both mean absolute error and Pearson correlation against human expert scores. As shown in **Table R2**, Gemini‑2.5‑Flash consistently achieves the **highest correlations (0.88-0.95)** across both IPhO 2025 and IPhO 2024, along with the **smallest mean absolute errors (0.41-0.74)**, providing strong evidence of its alignment with expert judgment. In comparison, GPT‑4o exhibits substantially lower correlation (0.41) and much larger error (2.41) on average. We have included these results in **Section 5.4** and **Appendix E.3**.
>
> **Table R2. Statistical Analysis of Different Graders.**
> ||Mean Absolute Error|||Pearson Correlation|||
> -|-|-|-|-|-|-
> Grader|Gemini-2.5-Flash|DeepSeek-R1|GPT-4o|Gemini-2.5-Flash|DeepSeek-R1|GPT-4o
> IPhO 2025|0.41|0.88|0.79|0.95|0.90|0.75
> IPhO 2024|0.74|1.60|4.02|0.88|0.74|0.78
> **All**|**0.58**|**1.24**|**2.41**|**0.95**|**0.86**|**0.41**

---

> ### Author Response · Authors · 2025-11-27
>
> **Comment 3. On the Impact of Step-Level Grading:** Your evaluation metric is max(answer-level score, step-level score). This is a sensible choice. Could you provide a brief quantitative analysis of how frequently the step-level score was the decisive factor (i.e., when a model got the final answer wrong but still received partial credit for correct steps)? This would provide powerful evidence for the necessity of your more fine-grained evaluation method.
>
> **Response 3.** We adopt the maximum of answer- and step-level scores to align with human grading logic: if the final answer is correct, full marks are awarded; if it is incorrect, partial credit is given based on intermediate steps. Therefore, we typically select the higher of the two scores as the final score per problem. This approach captures the benefits of step-level granularity while avoiding underestimation of correct final answers.
>
> To quantify the impact of step-level grading, we analyzed how frequently step-level scores exceeded answer-level scores. As shown in **Table R3**, across three inference runs on IPhO 2025, step-level scores were the dominant contributor in **25.6% to 46.2%** of the problems. This highlights the substantial contribution of step-level evaluation in fairly recognizing partial reasoning competence. Relying solely on answer-level scores can lead to underestimated performance, as incorrect answer extraction or mismatched yet equivalent final answers may obscure valid intermediate reasoning. More importantly, our scoring methodology is aligned with official Olympiad grading rubrics, which assign partial points for intermediate steps. This alignment ensures that model scores can be directly and meaningfully compared to human contestants under the same criteria. We haved included these discussion in **Section 5.4** and **Appendix E.2**.
>
> **Table R3. Frequency of Step-Level Score Dominance.**
> |IPhO 2025|Run 1|Run 2|Run 3|
> -|-|-|-
> Gemini-2.5-Pro|30.8%|30.8%|25.6%
> Gemini-2.5-Flash-Thinking|33.3%|41.0%|46.2%
> Claude-4-Sonnet-Thinking|41.0%|35.9%|35.9%
>
> ---
>
> **Comment 4. On Medal Threshold Determination:** For Olympiads with both theory and experimental parts (IPhO, EuPhO), you set the medal threshold based on the "lowest theoretical exam score" of human medalists. This seems reasonable, but could it be skewed by a contestant who scored extremely high on the experimental exam and just high enough on the theory exam to secure a medal? How sensitive are the medal thresholds to such outliers, and does this methodology create a robust and representative performance bar?
>
> **Response 4.** Thank you for the insightful question. Our use of the *lowest theoretical score among human medalists* is grounded in how Olympiads award medals. Since medals are determined by total score, a contestant must still meet a **minimum level of theoretical competence** to plausibly obtain any medal. Any real constestant scoring *below* this theory threshold could not have been a medalist. This makes the lowest theoretical medalist score a conservative and meaningful benchmark for evaluating AI's theoretical ability.
>
> To verify robustness, we examined whether these thresholds were distorted by outliers with unusually high experimental scores. The theory scores of medalists are tightly clustered. For IPhO 2025, the gap between the lowest and second-lowest theory scores is only **0.8** for gold, **0.7** for silver, and **0.4** for bronze, indicating no extreme cases. We further checked how often contestants above our adopted thresholds actually received medals in Table R4:
>
> **Table R4. Medal rates for contestants above theory thresholds.**
> |Olympiad|Medal|Theory Threshold|% Above Threshold Who Got Medal|
> |-|-|-|-
> |IPhO 2025|Gold|≥ 19.7|82.2%
> ||Silver+|≥ 12.1|77.1%
> ||Bronze+|≥ 7.2|83.6%
> |IPhO 2024|Gold|≥ 20.8|85.7%
> ||Silver+|≥ 11.1|85.5%
> ||Bronze+|≥ 3.6|82.2%
>
> These results show that contestants exceeding our theory thresholds overwhelmingly earn corresponding medals. The thresholds are therefore **robust and representative**, providing a reasonable bar for assessing whether an AI model demonstrates medal-level theoretical performance. The ideal approach is to include experimental performance and apply official medal thresholds based on total scores. We will make efforts in future work to support such comprehensive evaluations.

---

> ### Author Response · Authors · 2025-11-27
>
> **Comment 5. On Future Directions for Experimental Physics:** You correctly identify the inability to perform experiments as a key limitation. Beyond simply stating this, could you elaborate on what you believe are the most critical research steps needed to bridge this "embodiment" gap? For example, would this involve creating simulated physics environments, developing agents capable of active experimentation, or something else entirely? A more detailed perspective would add valuable depth to the future work discussion.
>
> **Response 5.** Bridging the gap between theoretical reasoning and experimental competence requires progress at multiple levels of capability and evaluation. We envision three complementary research directions. **(1) Experimental reasoning through multimodal inputs:** Experimental setups, instruments, and measurement data are provided as figures or tables; this tests whether models can understand procedures, interpret data, and reason about experiments without direct interaction. **(2) Simulated experimental environment:** Models operate within simulated physics labs and actively manipulate virtual apparatus; this requires interactive decision‑making and closed‑loop reasoning. **(3) Embodied experimentation:** Models interface with robotic embodiments such as manipulators or dexterous hands to perform real experiments in the physical world. This aligns with emerging VLA (Vision–Language–Action) paradigms and represents a long‑term frontier. Due to space limitations, we have added a brief discussion in **Section 5.5** and will continue to explore how future benchmarks can incorporate experimental physics evaluation.

---

### Official Review · Reviewer_zQZx · 2025-10-31

**Soundness:** 2
**Presentation:** 3
**Contribution:** 2
**Rating:** 2
**Confidence:** 4

**Summary:**

This paper introduces HiPhO, a benchmark for evaluating LLMs and MLLMs on recent high-school physics Olympiad problems. It compiles 13 exams from 2024–2025, covering both text and diagram-based questions, and applies official marking schemes for fine-grained, step-level grading. Model scores are further mapped to Olympiad medal thresholds, enabling direct comparison with human contestants. Experiments on 30 state-of-the-art models show that closed-source reasoning models reach near–gold-level performance, while most open-source MLLMs remain around bronze. The benchmark provides a systematic and human-aligned testbed for multimodal physical reasoning, though its contribution is primarily evaluative rather than methodological.

**Strengths:**

1. The benchmark includes recent Olympiad exams from 2024–2025, ensuring up-to-date coverage.
2. The dataset construction is technically thorough, with careful extraction, verification, and alignment to official marking schemes.
3. The multimodal categorization is well defined, distinguishing text-only, illustration, variable-based, and data-figure problems for fine-grained evaluation.

**Weaknesses:**

1. The paper primarily serves as a benchmark contribution, offering extensive evaluations but limited analytical depth. While the dataset and results are well organized, the work does not provide deeper investigation into model behavior or new insights into the reasoning mechanisms of LLMs, resulting in limited conceptual originality.
2. The introduction of the Mean Normalized Score (MNS) lacks sufficient explanation. The paper briefly presents the metric without clarifying its motivation, interpretation, or comparison to other normalization choices. As a result, it remains unclear how MNS meaningfully captures modality-specific reasoning performance beyond simple score rescaling.
3. While the paper acknowledges that experimental and embodied reasoning tasks are excluded, it still frames HiPhO as a benchmark for “human-level physical reasoning.” This phrasing may overstate the benchmark’s scope, since the evaluated skills are limited to theoretical problem solving rather than full-spectrum physical reasoning.
4. Although the paper introduces step-level scoring based on official marking schemes, it does not include any corresponding step-level analysis or discussion. Without examining how models perform across different reasoning steps, the potential advantages of this fine-grained evaluation are underutilized.

**Questions:**

1. What is the intended motivation behind introducing the Mean Normalized Score (MNS), and how should it be interpreted relative to standard scoring metrics?
2. Given that experimental and embodied reasoning tasks are excluded, would the authors consider refining how “human-level physical reasoning” is framed in the paper?
3. Have the authors analyzed model performance at the step level to identify specific reasoning weaknesses or error patterns? If not, could such analysis provide deeper insights into how models approach multi-step physics problems?

---

> ### Author Response · Authors · 2025-11-27
>
> Thank you for recognizing the strengths of our work in up-to-date coverage, careful dataset construction, and fine-grained multimodal categorization. **We have revised the paper accordingly, added detailed error analysis, and clarified the motivation behind our metrics.**
>
> ---
>
> **Comment 1.** The paper primarily serves as a benchmark contribution, offering extensive evaluations but limited analytical depth. While the dataset and results are well organized, the work does not provide deeper investigation into model behavior or new insights into the reasoning mechanisms of LLMs, resulting in limited conceptual originality.
>
> **Response 1.** In addition to extensive evaluations, our paper provided several analyses of model reasoning:
>
> 1. **Closed-source vs. Open-source** (Section 4.1): We examine disparities between closed- and open-source models.
> 2. **AI vs. Human** (Figure 5, Section 4.2): We analyze the performance gap between top-performing models and human contestants.
> 3. **Modality Sensitivity** (Section 5.1): We compare model performance across input modalities, identifying significant drops on **diagram-heavy items**, especially those requiring variable recognition and data interpretation.
> 4. **Physics Subfields** (Section 5.2): We evaluate performance across subfields, showing that **Optics** consistently poses greater challenges than Mechanics or Thermodynamics.
>
> To further examine model reasoning, we conduct a detailed **error analysis** in **Section 5.3** and introduce a taxonomy of **three main error types** and **nine fine-grained subtypes**. Unlike prior work that assigns a single error label per response, our benchmark applies **step-level evaluation**, allowing **multiple error tags** per solution. For instance, a failure to extract values from a diagram maps to *Data-reading Inaccuracy*, while a misstep in symbolic manipulation maps to *Faulty Symbolic Derivation*. This approach yields more **fine-grained and comprehensive error statistics**.
>
> As shown in **Figure 7** (Section 5.4), our analysis shows that most failures cluster in four key subtypes:
> - **Misapplication of Physical Laws**
> - **Lack of Physics Knowledge**
> - **Faulty Symbolic Derivation**
> - **Task Misunderstanding**
>
> These findings reflect the core challenges of Olympiad-level physics: deep conceptual reasoning, complex multi-step derivations, and nuanced understanding of problem context. Our enhanced analysis offers deeper insights into model limitations and supports future efforts toward improving scientific reasoning in LLMs.
>
> ---
>
> **Comment 2 (Question 1).** The introduction of the Mean Normalized Score (MNS) lacks sufficient explanation. The paper briefly presents the metric without clarifying its motivation, interpretation, or comparison to other normalization choices. As a result, it remains unclear how MNS meaningfully captures modality-specific reasoning performance beyond simple score rescaling. What is the intended motivation behind introducing the Mean Normalized Score (MNS), and how should it be interpreted relative to standard scoring metrics?
>
> **Response 2.**
> We introduced the Mean Normalized Score (MNS) to enable fairer cross-modality comparisons, particularly when problems vary in difficulty and total points. Two common normalization strategies are:
>
> - **Mean Normalized Score (MNS):**
>   $MNS = \frac{1}{N} \sum_{i=1}^{N} \frac{s_i}{t_i}$
>   where $s_i$ is the model's score and $t_i$ is the total score for problem $i$. This computes the **average of per-problem score rates**, assigning equal weight to each question regardless of its full mark.
>
> - **Rescaled Score:**
>   $RS = \frac{\sum_{i=1}^{N} s_i}{\sum_{i=1}^{N} t_i}$
>   This computes a **weighted average**, where problems with higher total points contribute more to the final score.
>
> In our benchmark, the total scores per problem differ significantly across Olympiads (e.g., the average per-question score is **0.75** for IPhO 2025 and **2.13** for PanPhO 2025). As a result, **Rescaled Score introduces bias**, giving greater weight to high-value questions and certain Olympiads. In contrast, **MNS avoids this bias** by evaluating each problem equally, conceptually similar to computing average accuracy. This makes MNS a more balanced and interpretable metric for evaluating modality-specific reasoning performance.

---

> ### Author Response · Authors · 2025-11-27
>
> **Comment 3 (Question 2).** While the paper acknowledges that experimental and embodied reasoning tasks are excluded, it still frames HiPhO as a benchmark for "human-level physical reasoning." This phrasing may overstate the benchmark's scope, since the evaluated skills are limited to theoretical problem solving rather than full-spectrum physical reasoning. Given that experimental and embodied reasoning tasks are excluded, would the authors consider refining how "human-level physical reasoning" is framed in the paper?
>
> **Response 3.** This comment appears to be based on a **misunderstanding**. We would like to clarify that we do **NOT** claim HiPhO to be a benchmark for "human-level physical reasoning." Rather, the paper explicitly states that HiPhO adopts a **human-aligned evaluation** strategy based on official marking schemes, which enables **human-level comparisons** with Olympiad contestants. These are two fundamentally different concepts.
>
> - **Human-Aligned Evaluation:** Based on official marking schemes, we evaluate model outputs using answer-level and step-level scoring to compute **exam scores** rather than accuracy, thereby aligning with how humans are graded.
> - **Human-Level Comparison:** These exam scores then allow for direct comparison with real contestant scores and official medal thresholds, enabling us to assess whether a model achieves medal-level performance on theoretical exams.
>
> The phrase *"human-level physical reasoning"* appears only in **Section 5.5** (new version), where we explicitly highlight the current limitations of MLLMs. We clarify that achieving true human-level physical reasoning would require advances in multimodality, generation, and embodiment. Additionally, in **Section 6**, we state that incorporating evaluations of experimental physics in future work would enable a more authentic comparison with human reasoning. **Therefore, we do NOT overstate the scope of the benchmark; rather, we present *human-level physical reasoning* as a model limitation and a direction for future work.**
>
> ---
>
> **Comment 4 (Question 3).** Although the paper introduces step-level scoring based on official marking schemes, it does not include any corresponding step-level analysis or discussion. Without examining how models perform across different reasoning steps, the potential advantages of this fine-grained evaluation are underutilized.
>
> **Response 4.** As recommended, we further performed a fine-grained error analysis on 85 problems from the HiPhO benchmark where both Gemini-2.5-Pro and GPT-5 scored $\leq$75% of the total points. For each step-level criterion marked incorrect, we assign one of nine error subtypes under three main categories:
> - **Concept & Computation Error:** These errors stem from missing conceptual understanding or simple numerical mistakes, including *Task Misunderstanding*, *Lack of Physics Knowledge*, and *Calculation Mistakes*.
> - **Physics Reasoning Error:** These involve incorrect symbolic or physical reasoning, including *Misapplication of Physical Laws*, *Faulty Symbolic Derivation*, and *Dimensional Analysis Lapses*.
> - **Visual Reasoning Error:** These arise from misinterpreting diagrams or data, including *Scene Misinterpretation*, *Data-reading Inaccuracy*, and *Variable Misunderstanding*.
>
> As shown in **Fig. 7** (**Section 5.3**), our findings show that:
>
> - **Open-source models make notably more errors, especially in physics knowledge and law application**, indicating substantial room for improvement in domain understanding.
> - **Closed-source models mainly struggle with task misinterpretation and symbolic derivation**, underscoring the complexity of Olympiad-style multi-step reasoning.
> - **Scene misinterpretation is the most frequent visual error for open-source models**, highlighting persistent challenges with interpreting physical setup in figures.
>
> These patterns expose key weaknesses in both physics and visual reasoning. **Representative examples of each error type are provided in Appendix F.**

---

### Official Review · Reviewer_sYtg · 2025-11-14

**Soundness:** 3
**Presentation:** 3
**Contribution:** 2
**Rating:** 4
**Confidence:** 4

**Summary:**

The paper introduces HiPhO, a human-aligned benchmark for high-school Physics Olympiads that aggregates 13 up-to-date 2024–2025 exams across international and regional competitions, evaluates with answer- and step-level scoring aligned to official marking schemes, and enables direct model–human comparison via exam scores and medal thresholds. A large-scale study over 30 SOTA (M)LLMs shows closed-source reasoning MLLMs frequently reach gold level yet still trail the best human contestants; open-source MLLMs mostly sit at bronze–silver, while several open-source LLMs (text-only) achieve multiple golds. Analyses reveal substantial drops on diagram-heavy items (variable/data figures) and consistent difficulty in Optics relative to other fields.

**Strengths:**

- There is a clear data construction and verification pipeline
- Large-scale evaluation across 30 (M)LLMs with per-modality and field analysis
- The paper is easy to follow. Dataset statistics, thresholds, and score distributions are explicit.
- The mixed modality taxonomy exposes where (M)LLMs reasoning falters.

**Weaknesses:**

- Given the recency (2024–2025 exams), many models might have trained on or memorized some content. The paper mentions up-to-date coverage but does not provide data leakage / contamination checks.
- Taking the maximum of answer- and step-level scores can inflate credit if the grader is permissive at either level. Consider reporting both components and their correlation, plus a conservative aggregate (e.g., min or weighted combination) as a sensitivity check.

**Questions:**

- The paper reports model performance in terms of Olympiad medal tiers (gold/silver/bronze), but there isn't a clear justification for **why** this human-referenced scale is necessary or preferable to standard metrics. Could the authors explain the decision-making or evaluation scenarios where medal-level reporting adds value beyond accuracy scores? Do the medal tiers correlate with deployment-relevant reliability?
- The paper claims that HiPhO is “the first benchmark to compare model performance with medal cutoffs and contestant scores” and introduces a medal-table ranking “just like an Olympiad.” However, this paper (https://arxiv.org/pdf/2406.16772) already proposed an Olympic medal table approach to rank AI models across subjects in mid-2024. Could the authors clarify how HiPhO’s contribution differs from OlympicArena’s earlier medal-table methodology?
- What is the inter-rater reliability between human experts and the automatic step-scorer across multiple exams? How did the authors ensure that using a strong judge model would not impose errors?

---

> ### Author Response · Authors · 2025-11-27
>
> Thank you for recognizing the strengths of our work, including the data construction and verification pipeline, large-scale evaluation, clear presentation, and the mixed-modality analysis. We have revised the paper accordingly.
>
> ---
>
> **Comment 1.** Given the recency (2024–2025 exams), many models might have trained on or memorized some content. The paper mentions up-to-date coverage but does not provide data leakage / contamination checks.
>
> **Response 1.** We took several measures to reduce the risk of data leakage. First, following OlympiadBench [1], we sourced problems directly from official websites and converted them from PDF to markdown, which helps prevent these items from appearing in model pre‑training corpora. Second, we intentionally focused on the most recent 2024–2025 Olympiads to mitigate temporal overlap with model training. For example, the official APhO 2025 problems were released on August 16, 2025, which is later than the release dates of all evaluated models.
>
> As suggested, we further conducted explicit data‑leakage checks. Following the **partial‑prompt completion** strategy in [2], we randomly sampled 90 problems (25% benchmark data), ensuring coverage across all Olympiads. For each problem, we provided the first 60% of the content and asked the model to greedily complete the remainder. We then computed the same metrics as in [2]:
>
> - **ROUGE-L (Partial-Prompt Completion Rate):** Measures the overlap of the longest common subsequence between the generated and reference text.
> - **ACC (Partial-Prompt Answer Accuracy):** Measures the proportion of problems where the generated text contains the correct answer.
>
> As shown in **Table R1**, we evaluated 10 top-performing models and found that their partial‑prompt completion rates are all below **15%**, with answer accuracy below **4%** on average. The detailed results in **Tables R2 & R3** further confirm that across all 13 Olympiads, most completion rates remain below **20%** and most answer accuracies remain at **0%**. These findings indicate that the HiPhO benchmark exhibits **minimal data leakage** and remains **highly challenging**. We have included these data leakage checks in **Section 3.4** and **Appendix B.3**.
>
> **Table R1. Summary of Data Leakage Check (%) on the HiPhO subset.**
> |Closed-source Models|ROUGE-L|ACC|Open-source Models|ROUGE-L|ACC|
> -|-|-|-|-|-
> Gemini-2.5-Pro|11.8|1.7|Intern-S1|10.9|2.2
> Gemini-2.5-Flash-Thinking|11.8|1.7|InternVL3-78B-Instruct|12.2|1.1
> GPT-5|12.1|3.7|Qwen2.5-VL-72B-Instruct|10.8|1.1
> Grok-4|14.9|1.7|DeepSeek-R1|10.3|0.0
> Claude-4-Sonnet-Thinking|10.7|1.1|Qwen3-235B-A22B|8.4|1.1
>
> ---
>
> **Table R2. Partial-prompt completion rate (\%) across all Olympiads.**
> |ROUGE-L|IPhO||APhO|EuPhO||NBPhO||PanPhO||PanMechanics||F=MA||
> |-|-|-|-|-|-|-|-|-|-|-|-|-|-|
> ||2025|2024|2025|2025|2024|2025|2024|2025|2024|2025|2024|2025|2024|
> |*Closed-source Models*||||||||||||||
> |Gemini-2.5-Pro|15.9|15.2|14.9|6.5|14.3|17.2|15.9|13.3|13.9|4.1|5.1|6.1|4.9|
> |Gemini-2.5-Flash-Thinking|11.5|11.4|10.0|13.5|18.5|7.2|7.6|12.6|11.2|6.9|7.4|16.2|19.3|
> |GPT-5|11.2|8.0|13.3|11.4|11.1|15.1|12.0|13.8|11.9|5.6|8.6|15.7|20.2|
> |Grok-4|12.2|10.9|13.4|15.0|22.1|15.7|14.9|16.0|13.1|5.1|10.3|18.2|29.3|
> |Claude-4-Sonnet-Thinking|11.5|14.7|12.9|10.5|7.1|12.3|11.5|13.5|13.6|3.7|5.9|8.4|7.3|
> |*Open-source Models*||||||||||||||
> |Intern-S1|12.9|13.6|13.4|9.1|12.7|11.0|13.7|13.4|12.5|2.5|3.5|7.9|10.0|
> |InternVL3-78B-Instruct|11.9|14.0|19.1|6.9|12.3|12.5|10.6|13.2|14.0|14.7|10.6|7.6|6.4|
> |Qwen2.5-VL-72B-Instruct|12.5|12.4|18.2|10.3|11.1|11.2|10.3|12.4|12.2|3.0|7.4|4.6|7.5|
> |DeepSeek-R1|13.0|13.3|12.0|6.6|12.5|13.1|10.5|10.2|13.1|1.3|5.7|8.2|9.8|
> |Qwen3-235B-A22B|12.0|13.8|10.9|7.9|9.7|8.0|8.3|7.7|11.5|0.0|1.5|6.5|5.8|
>
> ---
>
> **Table R3. Partial-prompt answer accuracy (\%) across all Olympiads.**
> |ACC|IPhO||APhO|EuPhO||NBPhO||PanPhO||PanMechanics||F=MA||
> |-|-|-|-|-|-|-|-|-|-|-|-|-|-|
> ||2025|2024|2025|2025|2024|2025|2024|2025|2024|2025|2024|2025|2024|
> |*Closed-source Models*|||||||||||||||
> |Gemini-2.5-Pro|0.0|0.0|0.0|0.0|8.3|0.0|0.0|9.1|0.0|0.0|0.0|0.0|0.0|
> |Gemini-2.5-Flash-Thinking|0.0|0.0|0.0|0.0|8.3|0.0|0.0|9.1|0.0|0.0|0.0|0.0|0.0|
> |GPT-5|11.1|0.0|12.5|0.0|0.0|0.0|0.0|12.1|0.0|0.0|0.0|0.0|0.0|
> |Grok-4|11.1|0.0|0.0|0.0|8.3|0.0|0.0|0.0|0.0|0.0|0.0|0.0|0.0|
> |Claude-4-Sonnet-Thinking|0.0|0.0|0.0|0.0|0.0|0.0|0.0|0.0|0.0|14.3|0.0|0.0|0.0|
> *Open-source Models*||||||||||||||
> |Intern-S1|11.1|0.0|0.0|0.0|0.0|0.0|0.0|0.0|0.0|14.3|0.0|0.0|0.0|
> |InternVL3-78B-Instruct|0.0|0.0|0.0|0.0|0.0|0.0|0.0|0.0|0.0|0.0|0.0|0.0|16.7|
> |Qwen2.5-VL-72B-Instruct|0.0|0.0|0.0|0.0|0.0|0.0|0.0|0.0|0.0|14.3|0.0|0.0|0.0|
> |DeepSeek-R1|0.0|0.0|0.0|0.0|0.0|0.0|0.0|0.0|0.0|0.0|0.0|0.0|0.0|
> |Qwen3-235B-A22B|11.1|0.0|0.0|0.0|0.0|0.0|0.0|0.0|0.0|0.0|0.0|0.0|0.0|
>
> [1] Olympiadbench: A challenging benchmark for promoting agi with olympiad-level bilingual multimodal scientific problems. ACL 2024.
>
> [2] Reasoning or memorization? unreliable results of reinforcement learning due to data contamination. arXiv 2507.

---

> ### Author Response · Authors · 2025-11-27
>
> **Comment 2.** Taking the maximum of answer- and step-level scores can inflate credit if the grader is permissive at either level. Consider reporting both components and their correlation, plus a conservative aggregate (e.g., min or weighted combination) as a sensitivity check.
>
> **Response 2.** As suggested, we conducted a sensitivity analysis using eight independent grading runs per model and added the analysis to **Appendix E.1**. In addition to reporting the answer- and step-level scores, we include their average and maximum. As shown in **Table R4**, all four scoring metrics exhibit strong internal consistency. The rankings based on maximum scores generally align with those based on answer- and step-level scores, indicating stable relative performance across metrics. Moreover, step-level scoring helps distinguish model performance more finely, as it awards partial credit for intermediate reasoning steps even when the final answer is incorrect.
>
> We adopt the maximum of answer- and step-level scores to align with human grading logic: if the final answer is correct, full marks are awarded; if it is incorrect, partial credit is given based on intermediate steps. Therefore, we typically select the higher of the two scores as the final score per problem. This approach captures the benefits of step-level granularity while avoiding underestimation of correct final answers.
>
> **Table R4. Sensitivity Analysis of Different Evaluation Metrics.**
> |Olympiad|IPhO 2025||||IPhO 2024||||
> |-|-|-|-|-|-|-|-|-|
> Metric|answer-level|step-level|average|**max**|answer-level|step-level|average|**max**
> Gemini-2.5-Pro|18.2|20.1|19.1|**22.7**|24.4|25.7|25.1|**25.9**
> Gemini-2.5-Flash-Thinking|16.1|19.2|17.6|**20.2**|20.1|23.7|21.9|**23.9**
> Claude-4-Sonnet-Thinking|16.0|17.3|16.7|**19.0**|20.2|21.5|20.8|**22.0**
>
> ---
>
> **Comment 3.** There isn't a clear justification for why this human-referenced scale is necessary or preferable to standard metrics. Could the authors explain the decision-making or evaluation scenarios where medal-level reporting adds value beyond accuracy scores? Do the medal tiers correlate with deployment-relevant reliability?
>
> **Response 3.** Our core objective is to assess the **gap between (M)LLMs and top-performing humans in Olympiad-level physics reasoning**. Given this focus, it is natural and necessary to adopt **human-referenced evaluation**, i.e., the official scoring schemes and medal thresholds, which allow us to compute exam scores rather than accuracy and enable **direct comparisons** between model performance and human contestants. Compared to standard accuracy, **medal-level exam score reporting** offers several unique advantages:
>
> 1. **Finer-grained evaluation**: Accuracy typically reflects only the correctness of the final answer. In contrast, our exam scores are computed using **official marking schemes**, which assign points for intermediate reasoning steps. This better captures the model's actual performance in real exams.
>
> 2. **Alignment with human performance**: Human-AI comparison is seen as a cornerstone of AGI evaluation [3], as highlighted by recent reports that Gemini with Deep Think achieved gold-medal performance at IMO 2025. Benchmarks such as GSM8K [4] (math), PHYBench [5] (physics), and GSO [6] (coding) emphasize human-level performance, but their expert baselines are typically self-defined. Physics Olympiads, by contrast, come with **real and official contestant scores**, where the "experts" are medal-winning students from around the world. This allows for **authentic, standardized comparisons** between model and human performance under the same grading criteria.
>
> 3. **Medals as performance tiers**: Medal thresholds represent **official, human-defined proficiency levels**. Mapping model scores to these tiers adds a **qualitative layer of interpretation** beyond raw numerical results. For example, a model reaching the gold medal threshold on IPhO 2025 is not simply achieving a high score; it is operating at the level of the **top ~10% of global human contestants**, demonstrating reasoning ability comparable to the world's top-performing students in physics.
>
> Medal tiers serve as a **valuable indicator of high-level reasoning competence** and can inform deployment decisions. For example, deploying an open-source model with gold-medal performance offers a good balance of capability and cost-effectiveness. As such, medal-based evaluation offers a more **intuitive, comparative, and aspirational framework**, particularly in the context of **AI-human alignment and AGI development**.
>
> [3] Olympiad-level formal mathematical reasoning with reinforcement learning. Nature 2025.
>
> [4] GSM-Plus: A Comprehensive Benchmark for Evaluating the Robustness of LLMs as Mathematical Problem Solvers. ACL 2024.
>
> [5] PHYBench: Holistic evaluation of physical perception and reasoning in large language models. arXiv 2504.
>
> [6] GSO: Challenging Software Optimization Tasks for Evaluating SWE-Agents. arXiv 2505.

---

> ### Author Response · Authors · 2025-11-27
>
> **Comment 4.** The paper claims that HiPhO is "the first benchmark to compare model performance with medal cutoffs and contestant scores" and introduces a medal-table ranking "just like an Olympiad." However, this paper already proposed an Olympic medal table approach to rank AI models across subjects in mid-2024. Could the authors clarify how HiPhO's contribution differs from OlympicArena's earlier medal-table methodology?
>
> **Response 4.** The contributions of HiPhO and OlympicArena are **fundamentally different**, particularly in their **medal definition**, **evaluation setting**, and **scoring metric**:
>
> 1. **Medal Definition**: HiPhO awards a gold medal only if the **exam score matches or exceeds the official gold threshold**, enabling an **AI-vs-human** comparison. In contrast, OlympicArena awards gold to models that **rank first among all AI models**, based solely on **AI-vs-AI** performance, with no reference to human scores.
>
> 2. **Evaluation Setting**: HiPhO evaluates models on **full Olympiad exam papers**, faithfully replicating real test conditions. OlympicArena, however, uses **selected problems from multiple exams**, which does not reflect full exam-level reasoning.
>
> 3. **Scoring Metric**: HiPhO uses official scoring schemes to compute **exam scores**, aligning directly with human grading. OlympicArena uses **accuracy**, which is not comparable to human scores or medal thresholds.
>
> **While both works use the term "medal table," their definitions, evaluation settings, and underlying goals are entirely distinct.** HiPhO's contribution lies in establishing a **human-aligned benchmark** grounded in official Olympiad standards, enabling direct and meaningful comparison between models and top-performing human contestants.
>
> ---
>
> **Comment 5.** What is the inter-rater reliability between human experts and the automatic step-scorer across multiple exams? How did the authors ensure that using a strong judge model would not impose errors?
>
> **Response 5.** While we do not claim to eliminate grading errors entirely, our empirical analysis shows that the widely used GPT‑4o grader introduces noticeable discrepancies compared with human experts, motivating our choice of Gemini-2.5-Flash as a more reliable alternative for HiPhO. To validate this choice, we compared three graders: Gemini‑2.5‑Flash (used in our work), GPT‑4o (widely adopted in prior studies [7,8,9]), and human experts, across both IPhO 2025 and IPhO 2024. For each of three evaluated models, we independently graded three inference runs using all three graders. As shown in **Table R5** (**Appendix E.3**), **Gemini‑2.5‑Flash consistently produces scores closely aligned with human experts (typically within 1 point)**. In contrast, GPT‑4o, whose lower competition scores make it less suitable for Olympiad-level evaluation, tends to significantly underestimate model performance and diverges more from human grading, especially on IPhO 2024.
>
> Although some variability naturally exists even among human judges, converting complex Olympiad marking schemes into consistent automatic scores remains an open challenge. We do not claim to resolve this entirely, but offer a demonstrably more aligned and practical alternative via Gemini‑2.5‑Flash. We hope future work will build on this direction to develop more principled and robust step-level grading systems for scientific reasoning.
>
> **Table R5. Comparison of Different Graders with Human Experts.**
> |Olympiad|IPhO 2025|||IPhO 2024|||
> |-|-|-|-|-|-|-|
> Grader|Gemini-2.5-Flash|GPT-4o|Human|Gemini-2.5-Flash|GPT-4o|Human
> Gemini-2.5-Pro|22.1|21.5|22.0|25.7|21.1|26.0
> Gemini-2.5-Flash-Thinking|19.9|20.4|19.1|23.5|20.0|23.3
> Claude-4-Sonnet-Thinking|19.3|18.5|19.0|22.1|17.4|21.3
>
> [7] UGPhysics: A Comprehensive Benchmark for Undergraduate Physics Reasoning with Large Language Models. ICML 2025.
>
> [8] PHYSICS: Benchmarking Foundation Models on University-Level Physics Problem Solving. arXiv 2503.
>
> [9] PhysicsArena: The First Multimodal Physics Reasoning Benchmark Exploring Variable, Process, and Solution Dimensions. arXiv 2505.

---

### Meta-Review · Area_Chair_fBGv · 2025-12-18

**Summary:**

Although the authors provided a lot of additional results during the rebuttal, several critical concerns remain.

Specifically, given that the primary contribution of this work is a new dataset, a rigorous assessment of data contamination and leakage is essential to ensure the validity of the benchmark results. The current rebuttal only offers preliminary findings based on partial-prompt completion tests using 25% of the benchmark data. A more exhaustive study employing varying ratios of partial prompts is necessary, consistent with established methodologies in the literature (e.g., [1]).

Furthermore, while the authors incorporated step-level analysis, grader reliability assessments, and fine-grained failure analysis in response to reviewer feedback, these additions appear to have been conducted in a rushed manner due to the constraints of the rebuttal period. A more comprehensive and systematic investigation is required to ensure that the conclusions drawn from these analyses are robust and reliable.

Overall, the paper currently sits marginally below the acceptance bar. I recommend that the authors improve these aspects of the work and submit the work to a future venue.

[1] Reasoning or memorization? unreliable results of reinforcement learning due to data contamination. arXiv 2507.

**Reviewer Concerns:**

The authors have partially addressed the following concerns via the rebuttal:

- Should use more conservative aggregation of answer- and step-level scores. Report individual scores and their correlation. [sYtg]
- Unjustified use of medal-level evaluation metrics. [sYtg, tc7S]
- Lack of discussion with related work like OlympicArena. [sYtg]
- How reliable are automatic evaluators compared to human experts? [sYtg]
- Lack of justification for the proposed metric Mean Normalized Score. [zQZx]
- Overstated scope. [zQZx]
- Lack of step-level analysis of discussion. [zQZx, tc7S]
- Lack of discussion on the limitation caused by using closed-source LLMs. [tc7S]
- Lack of fine-grained failure analysis. [FQst]

The following concerns are still outstanding:
- Missing data leakage/contaimination checks. [sYtg, FQst]
- Limited analytical depth. [zQZx]
- Large-sample analysis is required to justify the choice of grader. [tc7S]

**Reviewer Scores:**

As no reviewers responded in the rebuttal discussion despite the authors' efforts to partially address some of their concerns individually, it is probable that a full discussion would have led to an adjustment of their respective scores.

---

### Decision · Program_Chairs · 2026-01-26

Reject